# Faster Convergence of Local SGD for Over-Parameterized Models

**Tiancheng Qin**                                                    *tq6@illinois.edu*
*Department of Industrial and Systems Engineering, Coordinated Science Laboratory*
*University of Illinois at Urbana-Champaign*

**S. Rasoul Etesami**                                                *etesami1@illinois.edu*
*Department of Industrial and Systems Engineering, Coordinated Science Laboratory*
*University of Illinois at Urbana-Champaign*

**Cesár A. Uribe**                                                   *cauribe@rice.edu*
*Department of Electrical and Computer Engineering*
*Rice University*

**Reviewed on OpenReview:** *https://openreview.net/forum?id=VBAKc4DtZ1*

## Abstract

Modern machine learning architectures are often highly expressive. They are usually over-parameterized and can interpolate the data by driving the empirical loss close to zero. We analyze the convergence of Local SGD (or FedAvg) for such over-parameterized models in the heterogeneous data setting and improve upon the existing literature by establishing the following convergence rates. For general convex loss functions, we establish an error bound of $\mathcal{O}(1/T)$ under a mild data similarity assumption and an error bound of $\mathcal{O}(K/T)$ otherwise, where $K$ is the number of local steps and $T$ is the total number of iterations. For non-convex loss functions we prove an error bound of $\mathcal{O}(K/T)$. These bounds improve upon the best previous bound of $\mathcal{O}(1/\sqrt{nT})$ in both cases, where $n$ is the number of nodes, when no assumption on the model being over-parameterized is made. We complete our results by providing problem instances in which our established convergence rates are tight to a constant factor with a reasonably small stepsize. Finally, we validate our theoretical results by performing large-scale numerical experiments that reveal the convergence behavior of Local SGD for practical over-parameterized deep learning models, in which the $\mathcal{O}(1/T)$ convergence rate of Local SGD is clearly shown.

## 1 Introduction

Distributed optimization methods have become increasingly popular in modern machine learning, owing to the data privacy/ownership issues and the scalability of learning models concerning massive datasets. The large datasets often make training the model and storing the data in a centralized way almost infeasible. That mandates the use of distributed optimization methods for training machine learning models. However, a critical challenge in distributed optimization is to reduce the communication cost among the local nodes, which has been reported as a major bottleneck in training many large-scale deep learning models (Zhang et al., 2017; Lin et al., 2017).

One naive approach to tackling this challenge is using the Minibatch Stochastic Gradient Descent (SGD) algorithm, which generalizes SGD to the distributed optimization setting by averaging the stochastic gradient steps computed at each node (or client) to update the model on the central server. Minibatch SGD has been shown to perform well in a variety of applications, see, e.g., Dekel et al. (2012); Cotter et al. (2011). Recently, Local SGD (Stich, 2018; Mangasarian, 1995) (also known as Federated Averaging) has attracted significant attention as an appealing alternative to Minibatch SGD to reduce communication cost, where

during a communication round, several local SGD iterations are performed at each node before the central server computes the average.

Local SGD has been widely applied in Federated Learning (Li et al., 2020), and other large-scale optimization problems and has shown outstanding performance in both simulation results (McMahan et al., 2017) as well as real-world applications such as keyboard prediction (Hard et al., 2018). At the same time, recent works have studied the theoretical convergence guarantees of Local SGD in various settings (Li et al., 2019; Koloskova et al., 2020; Gorbunov et al., 2021; Qin et al., 2020; Yang et al., 2021). Specifically, an $\mathcal{O}(\frac{1}{nT})$ convergence rate was shown for strongly convex loss functions (Karimireddy et al., 2020), where $n$ is the number of nodes and $T$ is the total number of iterations. Moreover, an $\mathcal{O}(\frac{1}{\sqrt{nT}})$ convergence rate was shown for general convex loss functions in Khaled et al. (2020). In addition, an $\mathcal{O}(\frac{1}{\sqrt{nT}})$ convergence rate was shown for non-convex loss functions (Yu et al., 2019; Haddadpour & Mahdavi, 2019). When the Polyak-Lojasiewicz condition is assumed, Haddadpour & Mahdavi (2019) showed $\mathcal{O}(\frac{1}{nT})$ convergence rate for non-convex loss functions and Maralappanavar et al. (2022) showed $\mathcal{O}(\exp(-T/K^2))$ convergence rate, where $K$ is the number of local steps.[1] These works made substantial progress toward understanding the theoretical convergence properties of the Local SGD. Their results are for general models without the over-parameterization (or interpolation) assumption.

However, despite past efforts, the current results have shortcomings in explaining the faster convergence of Local SGD compared to Minibatch SGD, which is significant especially when training large-scale deep learning models (McMahan et al., 2017). In Woodworth et al. (2020b), the authors give a lower bound on the performance of local SGD that is worse than the Minibatch SGD guarantee in the i.i.d. data setting (i.e., when all local loss functions are identical). The situation is even worse in the heterogeneous data setting (i.e., when local loss functions are different), which is the setting that we consider in this paper. Local SGD is shown to suffer from "client drift", resulting in unstable and slow convergence (Karimireddy et al., 2020), and it is known that Minibatch SGD dominates all existing analyses of Local SGD. (Woodworth et al., 2020a).

On the other hand, a key observation for explaining the fast convergence of SGD in modern machine learning was made by Ma et al. (2018) that says modern machine learning architectures are often highly expressive and are over-parameterized. Based on both theoretical and empirical evidence (Zhang et al., 2021; Chaudhari et al., 2019), most or all local minima in such over-parametrized settings are also global. Therefore, the authors in Ma et al. (2018) assumed *interpolation* of the data: the empirical loss at every data point can be driven to zero. Under such interpolation assumption, a faster convergence rate of SGD was proven (Ma et al., 2018; Vaswani et al., 2019). Furthermore, it was shown in Ma et al. (2018) that under certain conditions, a mini-batch size larger than some threshold $m^*$ is essentially helpless for SGD. This is important since, in distributed optimization, it means: *for Minibatch SGD, larger batch sizes will not speed up convergence, while for Local SGD, more local steps can potentially speed up convergence.* This provides a new direction for explaining the fast convergence of Local SGD for large-scale optimization problems as well as its faster convergence compared to Minibatch SGD.

Motivated by the above studies, in this paper, we formally study the theoretical convergence guarantees of Local SGD for training over-parameterized models in the heterogeneous data setting. Our results improve the existing literature and include the natural case of training large-scale deep learning models.

## 1.1 Related Works

Adopting a Neural Tangent Kernel (NTK) framework of analysis, two recent works, Huang et al. (2021); Deng et al. (2022) studied the convergence rate of Local SGD for specific over-parameterized Neural Networks and showed error bounds that are $\mathcal{O}(\exp(-T/K^2))$ and $\mathcal{O}(\exp(-T/K))$ respectively. However, both works focus on very restrictive and somewhat unrealistic types of Neural Networks. Huang et al. (2021) only considered two-layer fully connected Neural Networks with ReLU activation, and they require the width of the Neural Network to be $\Omega(N^4)$, where $N$ denotes the total number of data samples in the training set[2], which is not very realistic in practical applications. Likewise, Deng et al. (2022) considered fully connected Neural

---

[1]The PL condition is a generalization of strong convexity and requires the loss function to exhibit quadratic growth, which is a very strong assumption.

[2]This parameter $N$ is written as $n$ in the original work Huang et al. (2021).

Table 1: Existing theoretical bounds for local SGD for heterogeneous data. GC and NC stand for general convex and non-convex, $n$ is the number of nodes, $T$ is the number of total iterations, and $K$ is the number of local steps ($R = T/K$ is the number of communication rounds).

| Objective | Convergence Rate | Over-parameterzied | Extra Assumption | References |
|-----------|------------------|--------------------|------------------|------------|
| GC | $\mathcal{O}(\frac{1}{\sqrt{nT}})$ | No | / | Khaled et al. (2020) |
| NC | $\mathcal{O}(\frac{1}{\sqrt{nT}})$ | No | / | Koloskova et al. (2020) |
| NC | $\mathcal{O}(\frac{1}{nT})$ | No | PL condition | Haddadpour & Mahdavi (2019) |
| NC | $\mathcal{O}(\exp(-T/K^2))$ | Yes | PL condition | Maralappanavar et al. (2022) |
| **GC** | $\mathcal{O}(1/T)$ | **Yes** | $c > 0$ in (5) | **This work** |
| **GC** | $\mathcal{O}(K/T)$ | **Yes** | / | **This work** |
| **NC** | $\mathcal{O}(K/T)$ | **Yes** | / | **This work** |

Networks with ReLU activation but with multiple layers, and they require the width of the Neural Network to be $\Omega(N^{16})$, which is not practical in large-scale problems. As a comparison, we give analysis under the over-parameterized regime for strongly convex, convex, and non-convex loss functions that include the natural case of training large-scale Neural Networks but is not limited to it, which is a much broader analysis.

The work Li et al. (2022) also studied the convergence of Local SGD for over-parameterized Neural Networks. Utilizing the *no critical point* property of extra-wide Neural Networks shown in Allen-Zhu et al. (2019), they relaxed the commonly seen $L$-smoothness assumption of the local functions and proved the convergence of Local SGD but did not show an explicit convergence rate. Employing a new notion called *iterate bias*, Glasgow et al. (2022) recently showed lower bounds for the convergence rate of Local SGD without the over-parameterized assumption that matches (or nearly matches) the existing upper bounds, showing that without the over-parameterized assumption, the existing upper bound analysis is not improvable.

## 1.2 Contributions and Organization

Our main contributions can be summarized as follows:

- For general convex loss functions, we establish an error bound of $\mathcal{O}(1/T)$ under a mild data similarity assumption and an error bound of $\mathcal{O}(K/T)$, otherwise. Before our work, Zhang & Li (2021) showed the asymptotic convergence of Local Gradient Descent (GD) in this setting but did not provide an explicit convergence rate. To the best of our knowledge, the best convergence rate in this setting was $\mathcal{O}(1/\sqrt{nT})$ (Khaled et al., 2020) which was achieved without assuming the model being over-parametreized.

- For nonconvex loss functions, we prove an error bound of $\mathcal{O}(K/T)$. To the best of our knowledge, the best convergence rate in this setting was $\mathcal{O}(1/\sqrt{nT})$ (Koloskova et al., 2020) which was achieved without assuming the model being over-parametreized.

- We provide two problem instances to show that our convergence rates for the case of general convex and nonconvex functions are tight up to a constant factor under a reasonably small stepsize scheme.

- we validate our theoretical results by performing large-scale numerical experiments that reveal the convergence behavior of Local SGD for practical over-parameterized deep learning models, in which the $\mathcal{O}(1/T)$ convergence rate of Local SGD is clearly shown.

In fact, by establishing the above error bounds, we partially prove the effectiveness of local steps in speeding up the convergence of Local SGD, thus partially explaining the fast convergence of Local SGD (especially when compared to Minibatch SGD) when training large-scale deep learning models.

Our analysis builds upon the techniques used in Ma et al. (2018) and Vaswani et al. (2019) for analyzing centralized SGD in the over-parameterized setting and applies them to analyze both the local descent progress

and the global descent progress in Local SGD. Specifically, we adopt new techniques in the proof of Theorem 1 that directly relate the local progress with the global progress instead of measuring the progress made by $\bar{\mathbf{x}}^t$, where we made use of the consensus error, i.e., $\frac{1}{n}\sum_{i=1}^{n}\mathbb{E}\|\mathbf{x}_i^t - \bar{\mathbf{x}}^t\|^2$ to *improve* convergence, which is in contrast to prior works. The new technique allows us to use a constant stepsize that does not scale with $\frac{1}{K}$ and to establish better bounds. In the proof of Theorem 2, we use the techniques in Ma et al. (2018) and Vaswani et al. (2019) to bound both the global descent progress and the consensus error of Local SGD. These techniques may be of independent interest to the readers.

In Section 2, we formally introduce the problem. In Section 3, we state our main convergence results for general convex and non-convex local functions. We also provide a lower bound to show the tightness of our convergence rate bounds for reasonably small step sizes. We justify our theoretical bounds through extensive numerical results in Section 4. Conclusions are given in Section 5. We defer all the proofs to Section A.

## 2  Problem Formulation

We consider the problem of $n$ nodes $[n] = \{1, 2, \ldots, n\}$ that collaboratively want to learn an over-parameterized model with decentralized data as the following distributed stochastic optimization problem:

$$\min_{\mathbf{x} \in \mathbb{R}^d} \ f(\mathbf{x}) := \frac{1}{n}\sum_{i=1}^{n} f_i(\mathbf{x}), \tag{1}$$

where the function $f_i(\mathbf{x}) \triangleq \mathbb{E}_{\xi_i \sim \mathcal{D}_i} f_i(\mathbf{x}, \xi_i)$ denotes the local loss function, $\xi_i$ is a stochastic sample that node $i$ has access to, and $\mathcal{D}_i$ denotes the local data distribution over the sample space $\Omega_i$ of node $i$.

**Assumption 1** (Bounded below, $L$-smooth, unbiased gradient). *We assume $f(\mathbf{x})$ is bounded below by $f^\star$ (i.e., a global minimum exists), $f_i(\mathbf{x}, \xi_i)$ is $L$-smooth for every $i \in [n]$, and $\nabla f_i(\mathbf{x}, \xi_i)$ is an unbiased stochastic gradient of $f_i(\mathbf{x})$.*

Moreover, for some of our results, we will require functions $f_i(\mathbf{x}, \xi_i)$ to be $\mu$-strongly convex with respect to the parameter $\mathbf{x}$ as defined next.

**Assumption 2** ($\mu$-strong convexity). *There exists a constant $\mu \geq 0$, such that for any $\mathbf{x}, \mathbf{y} \in \mathbb{R}^d, i \in [n]$, and $\xi_i \in \Omega_i$, we have*

$$f_i(\mathbf{x}, \xi_i) \geq f_i(\mathbf{y}, \xi_i) + \langle \nabla f_i(\mathbf{y}, \xi_i), \mathbf{x} - \mathbf{y} \rangle + \frac{\mu}{2}\|\mathbf{x} - \mathbf{y}\|^2. \tag{2}$$

*If $\mu = 0$, we simply say that each $f_i$ is convex.*

The over-parameterized setting, i.e., when the model can *interpolate* the data completely such that the loss at every data point is minimized simultaneously (usually means zero empirical loss), can be characterized by the following two assumptions (Ma et al., 2018; Vaswani et al., 2019):

**Assumption 3** (Interpolation). *Let $\mathbf{x}^\star \in \arg\min_{x \in \mathbb{R}^d} f(\mathbf{x})$. Then, $\nabla f_i(\mathbf{x}^\star, \xi_i) = 0$, $\forall i \in [n]$, $\xi_i \in \Omega_i$.*

**Assumption 4** (Strong Growth Condition (SGC)). *There exists constant $\rho$ such that $\forall \mathbf{x} \in \mathbb{R}^d$, $i \in [n]$,*

$$\mathbb{E}_{\xi_i \sim \mathcal{D}_i}\|\nabla f_i(\mathbf{x}, \xi_i)\|^2 \leq \rho\|\nabla f(\mathbf{x})\|^2. \tag{3}$$

Notice that for the functions to satisfy SGC, local gradients at every data point must all be zero at the optimum $x^\star$. Thus, SGC is a stronger assumption than interpolation, which means Assumption 3 implies Assumption 2.

The SGC assumption can be viewed as an adaptation of a mild assumption, *Strong Growth with noise*, i.e.,

$$\mathbb{E}_{\xi_i \sim \mathcal{D}_i}\|\nabla f_i(\mathbf{x}, \xi_i)\|^2 \leq \rho\|\nabla f(\mathbf{x})\|^2 + \sigma^2, \tag{4}$$

to the over-parameterized/interpolation setting, which implies that the gradient with respect to each point converges to zero at the optimum, suggesting that $\sigma = 0$ in (4). The Strong Growth with noise assumption is

a generalization of the *Bounded Variance* assumption commonly used in the stochastic approximation setting, i.e.,

$$\mathbb{E}_{\xi_i \sim \mathcal{D}_i} \|\nabla f_i(\mathbf{x}, \xi_i)\|^2 \le \|\nabla f(\mathbf{x})\|^2 + \sigma^2.$$

The work Vaswani et al. (2019) discusses functions satisfying Assumption 4 (SGC) and shows that for linearly separable data, the squared hinge loss satisfies the assumption. In addition to that, we perform experimental verification in Appendix B using the same problem setup as in Section 4.1 (training over-parameterized ResNet18 Neural Network on the Cifar10 dataset) and show that Assumption 4 (SGC) is indeed a valid assumption for over-parameterized models in practice.

When the local loss functions are convex, we define the following quantity $c \in [0, 1]$ that allows us to measure the dissimilarity among them.

**Definition 1.** *Let Assumption 1, Assumption 2 and Assumption 3 (Interpolation) hold with $\mu \ge 0$. Let $\mathbf{x}^* \in \arg\min_{x \in \mathbb{R}^d} f(\mathbf{x})$. We define $c$ as the largest real number such that for all $\mathbf{x}_1, \dots \mathbf{x}_n \in \mathbb{R}^d$ and $\bar{\mathbf{x}} := \frac{1}{n} \sum_{i=1}^n \mathbf{x}_i$, we have*

$$\frac{1}{n} \sum_{i=1}^n (f_i(\mathbf{x}_i) - f_i(\mathbf{x}^*)) \ge c(f(\bar{\mathbf{x}}) - f(\mathbf{x}^*)). \tag{5}$$

If Assumption 2 and Assumption 3 (Interpolation) hold, the left hand side of (5) is always non-negative, which implies $c \ge 0$. In particular, by taking $\mathbf{x}_1 = \dots = \mathbf{x}_n$ we have $c \le 1$. Moreover, as the local loss functions become more similar, $c$ will become closer to 1. In particular, in the case of homogeneous local loss functions, i.e., $f_i = f \ \forall i$, using Jensen's inequality we have $c = 1$.

In the next section, we will proceed to establish our main convergence rate results for various settings of strongly convex, convex, and nonconvex local functions.

## 3 Convergence of Local SGD

This section reviews Local SGD and then analyzes its convergence rate under the over-parameterized setting.

In Local SGD, each node performs local gradient steps, and after every $K$ steps, sends the latest model to the central server. The server then computes the average of all nodes' parameters and broadcasts the averaged model to all nodes. Let $T$ be the total number of iterations in the algorithm. There is a set of communication times $\mathcal{I} = \{0, K, 2K, \dots, T = RK\}$[3], and in every iteration $t$, Local SGD does the following: i) each node performs stochastic gradient updates locally based on $\nabla f_i(\mathbf{x}, \xi_i)$, which is an unbiased estimation of $\nabla f_i(\mathbf{x})$, and ii) if $t$ is a communication time, i.e., $t \in \mathcal{I}$, it sends the current model to the central server and receives the average of all nodes' models. The pseudo-code for the Local SGD algorithm is provided in Algorithm 1.

### 3.1 Convergence Rate Analysis

We now state our main result on the convergence rate of Local SGD under over-parameterized settings for general convex functions.

**Theorem 1** (General convex functions). *Let Assumption 1, Assumption 2 and Assumption 3 (Interpolation) hold with $\mu = 0$, and let $c$ be defined as in Definition 1. Moreover, let*

$$w_t = \begin{cases} 1 & \text{if } t \in \mathcal{I} \text{ or } t + 1 \in \mathcal{I}, \\ c & \text{otherwise}, \end{cases}$$

*and define $W = \sum_{t=0}^{T-1} w_t$ and $\hat{\mathbf{x}}^T \triangleq \frac{1}{W} \sum_{i=0}^{T-1} w_t \bar{\mathbf{x}}^{(t)}$. If we follow Algorithm 1 with stepsize $\eta \le \frac{1}{2L}$ and $K \ge 2$, then*

$$\mathbb{E}[f(\hat{\mathbf{x}}^T) - f^*] \le \frac{K \|\mathbf{x}^{(0)} - \mathbf{x}^*\|^2}{\eta(cKT + 2(1-c)T)}.$$

---

[3]To simplify the analysis, we assume without loss of generality that $T$ is divisible by $K$, i.e., $T = RK$ for some $R \in \mathbb{N}$.

---

**Algorithm 1** Local SGD

---

1: **Input:** $\mathbf{x}_i^{(0)} = \mathbf{x}^{(0)}$ for $i \in [n]$, number of iterations $T$, the stepsize $\eta$, the set of communication times $\mathcal{I}$.
2: **for** $t = 0, \ldots, T-1$ **do**
3:     **for** $i = 1, \ldots, n$ **do**
4:         Sample $\xi_i^{(t)}$, compute $\nabla f_i(\mathbf{x}_i^{(t)}, \xi_i^{(t)})$
5:         $\mathbf{x}_i^{(t+\frac{1}{2})} = \mathbf{x}_i^{(t)} - \eta \nabla f_i(\mathbf{x}_i^{(t)}, \xi_i^{(t)})$
6:         **if** $t + 1 \in \mathcal{I}$ **then**
7:             $\mathbf{x}_i^{(t+1)} = \frac{1}{n} \sum_{j=1}^n \mathbf{x}_j^{(t+\frac{1}{2})}$
8:         **else**
9:             $\mathbf{x}_i^{(t+1)} = \mathbf{x}_i^{(t+\frac{1}{2})}$
10:         **end if**
11:     **end for**
12: **end for**

---

As a special case, if we choose $\eta = \frac{1}{2L}$, we have

$$\mathbb{E}[f(\hat{\mathbf{x}}^T) - f^*] \leq \frac{2KL\|\mathbf{x}^{(0)} - \mathbf{x}^*\|^2}{cKT + 2(1-c)T}. \tag{6}$$

The convergence of Local GD for general convex loss functions in the over-parameterized setting was shown earlier in Zhang & Li (2021) without giving an explicit convergence rate.[4] Instead, for similarity parameters $c > 0$ and $c = 0$, we give convergence rates of $\mathcal{O}(1/T)$ and $\mathcal{O}(K/T)$ for Local SGD, respectively. The significant difference between the convergence rates for the case of $c > 0$ and $c = 0$ suggests that having slight similarity in the local loss functions is critical to the performance of Local SGD, which also complies with the simulation findings in McMahan et al. (2017). To the best of our knowledge, Theorem 1 provides the first $\mathcal{O}(1/T)$ or $\mathcal{O}(K/T)$ convergence rates for Local SGD for general convex loss functions in the over-parameterized setting. On the other hand, in Section 3.2, we provide a problem instance suggesting that in the worst case, the $\mathcal{O}(K/T)$ convergence rate obtained here might be tight up to a constant factor.

It is worth noting that the speedup effect of local steps when $c > 0$ is a direct consequence of the $\mathcal{O}(1/T)$ convergence rate shown in Theorem 1. When $c = 0$, a closer look at (6) and the weights $w_t$ reveals that $w_t = 1$ if $t \in \mathcal{I}$ or $t + 1 \in \mathcal{I}$, implying that at least the first and the last local steps during each communication round is "effective". This, in turn, shows that local steps can speed up the convergence of Local SGD by at least a factor of 2[5].

For the case of non-convex loss functions, we have the following result.

**Theorem 2** (Non-convex functions). *Let Assumption 1, Assumption 4 (SGC) hold. If we follow Algorithm 1 with stepsize $\eta \leq \frac{1}{3KL\rho}$, and $K \geq 2$, we will have*

$$\min_{0 \leq t \leq T-1} \mathbb{E}\|\nabla f(\bar{\mathbf{x}}^t)\|^2 \leq \frac{9(f(\mathbf{x}_0) - f^*)}{\eta T}.$$

As a special case, if we choose $\eta = \frac{1}{3KL\rho}$, we have

$$\min_{0 \leq t \leq T-1} \mathbb{E}\|\nabla f(\bar{\mathbf{x}}^t)\|^2 \leq \frac{27KL\rho(f(\mathbf{x}_0) - f^*)}{T}. \tag{7}$$

---

[4]In fact, a convergence rate of $\mathcal{O}(1/\sqrt{T})$ was discussed in Zhang & Li (2021). However, the argument in their proof seems to have some inconsistencies. For more detail, please see Section D.

[5]Similar to Woodworth et al. (2020a), we compare the convergence rate of Local SGD to Minibatch SGD with $R = T/K$ steps and a batch size $K$ times larger than that of Local SGD. The convergence rate of Minibatch SGD (for over-parameterized setting), as stated in Theorem 6 in Vaswani et al. (2019), is $\frac{4LK(1+\rho)\|\mathbf{x}^{(0)} - \mathbf{x}^*\|^2}{T}$, which is at least 4 times slower than our rate when $c = 0$. On the other hand, according to our analysis and using Lemma 2, we can show the convergence rate of Minibatch SGD as $\frac{2LK\|\mathbf{x}^{(0)} - \mathbf{x}^*\|^2}{T}$, which is 2 times slower than Local SGD.

Theorem 2 provides an $\mathcal{O}(K/T)$ convergence rate for Local SGD for non-convex loss functions in the over-parameterized setting, which is the first $\mathcal{O}(1/T)$ convergence rate for Local SGD under this setting. However, this rate is somewhat disappointing as it suggests that local steps may not help the algorithm to converge faster. This is mainly caused by the choice of stepsize $\eta = \frac{1}{3KL\rho}$, which is proportional to $1/K$. On the other hand, in Section 3.2, we argue that this choice of stepsize may be inevitable in the worst case because there are instances for which the choice of stepsize $\eta$ greater than $\mathcal{O}(1/K)$ results in divergence of the algorithm.

### 3.2   Lower Bounds for the Convergence Rate of Local SGD

In this section, we present two instances of Problem (1) showing that the convergence rates shown in Section 3.1 are indeed tight up to a constant factor. First of all, we restrict to the scenario when Local SGD is run with stepsize $\eta \leq \frac{1}{L}$, as it is known from Nesterov et al. (2018) that Gradient Descent can provably diverge for stepsize $\eta > \frac{1}{L}$[6]. Then, we show that when Local SGD is run with stepsize $\eta \leq \frac{1}{L}$ and under the over-parameterized regime:

1. for general convex loss functions, there exist functions $f_i$ satisfying Assumption 1, Assumption 2 and Assumption 3 (Interpolation) with $\mu = 0$ and $c = 0$ in Definition 1, such that Local SGD incurs an error bound of $f(\bar{\mathbf{x}}^T) - f^* = \Omega(KL/T)$.

2. for non-convex loss functions, there exist functions $f_i$ satisfying Assumption 1, Assumption 4 (SGC), such that Local SGD with a stepsize $\eta \geq \frac{2}{LK}$ will not converge to a first-order stationary point.

**Proposition 1** (General Convex Functions). *There exists an instance of general convex loss functions $f_i$ satisfying Assumption 1, Assumption 2 and Assumption 3 (Interpolation) with $\mu = 0$ and $c = 0$ in Definition 1, such that Local SGD incurs an error bound of $f(\bar{\mathbf{x}}^T) - f^* = \Omega(KL/T)$.*

*Proof.* Consider Problem (1) in the following setting. Let $n = 4R = 4T/K$, $d = 1$, and $f_1(x) = \frac{L}{2}x^2$, $f_2(x) = f_3(x) = \cdots = f_n(x) = 0$. Then $f(x) = \frac{L}{2n}x^2$, and clearly every $f_i$ is $L$-smooth and satisfies Assumption 2 and Assumption 3 (Interpolation) with $\mu = 0$, $c = 0$. Suppose Algorithm 1 is run with stepsize $\eta \leq \frac{1}{L}$, and initialized at $\mathbf{x}^0 = 1$. We will show that $\min_{t \in [T]}(f(\bar{\mathbf{x}}^t) - f^*) \geq \frac{KL}{16T}$. To that end, first we note that the global optimal point is $\mathbf{x}^* = 0$, and local gradient steps for all nodes except node 1 keeps local variable unchanged. Moreover, since $\eta \leq 1/L$, we have $\mathbf{x}_1^t \in [0, 1]$, $\forall t \in [T]$. Therefore, $\{\bar{\mathbf{x}}^t\}$ is a non-increasing sequence that lies in interval $[0, 1]$. Thus, we only need to show $f(\bar{\mathbf{x}}^T) \geq \frac{KL}{16T}$.

Next, we claim that $\bar{\mathbf{x}}^{(r+1)K} \geq \frac{n-1}{n}\bar{\mathbf{x}}^{rK}$, $\forall r$. In fact, since $\mathbf{x}_1^{(r+1)K-1/2} \geq 0$ and $\mathbf{x}_i^{(r+1)K-1/2} = \bar{\mathbf{x}}^{rK}$, for $i = 2, \ldots, n$, we have

$$\bar{\mathbf{x}}^{(r+1)K} = \frac{1}{n}\sum_{i=1}^{n}\mathbf{x}_i^{(r+1)K-1/2} \geq \frac{n-1}{n}\bar{\mathbf{x}}^{rK}.$$

Therefore, we can write

$$f(\bar{\mathbf{x}}^T) = \frac{L}{2n}(\bar{\mathbf{x}}^T)^2 \geq \frac{L}{2n}(\frac{n-1}{n})^{2R} \geq \frac{L}{2n}(1 - \frac{2R}{n}) = \frac{L}{16R} = \frac{KL}{16T},$$

which completes the proof. $\square$

**Proposition 2** (Non-convex Functions). *There exists an instance of nonconvex loss functions $f_i$ satisfying Assumption 1, Assumption 4 (SGC), such that Local SGD with a stepsize $\eta \geq \frac{2}{LK}$ will not converge to a first-order stationary point.*

*Proof.* Consider Problem (1) in the following setting. Let $n = 2$, $d = 1$ and $f_1(x) = \frac{L}{2}x^2$, $f_2(x) = -\frac{L}{4}x^2$. Then $f(x) = \frac{L}{4}x^2$, and clearly every $f_i$ is $L$-smooth and satisfies Assumptions 4 (SGC) with $\rho = 2$. Suppose

---

[6]We note that $\eta \leq 1/L$ is a standard requirement when applying SGD-like algorithms on $L$-smooth functions, see e.g., Bubeck (2014). Many numerical experiments also show that stepsize $\eta > 1/L$ will cause divergence. In other words, when Local SGD is run with stepsize $\eta > \frac{1}{L}$, there are problem instances that Local SGD performs poorly

Algorithm 1 is run with stepsize $\eta \leq 1/L$ and initialized at $\mathbf{x}^0 = 1$. We want to show that for such distributed stochastic optimization problem, if we run Algorithm 1 for any stepsize $\eta \geq \frac{2}{LK}$, the gradient norm at any iterate will be lower bounded by $\min_{t \in [T]} \|\nabla f(\bar{\mathbf{x}}^t)\|^2 \geq \frac{L^2}{16}$.

First, we note that the global optimal point is $\mathbf{x}^* = 0$, which is the only critical point. Since $\eta \leq 1/L$, we have $\mathbf{x}_1^t \geq 0$, $\forall t \in [T]$, and local gradient steps for node 2 will always increase the value of $\mathbf{x}_2^t$. Next, we claim that if $\eta \geq \frac{2}{LK}$, then $\bar{\mathbf{x}}^{rK} \geq 1$, $\forall r$, and prove it by induction. First notice that $\bar{\mathbf{x}}^0 = 1 \geq 1$. Suppose $\bar{\mathbf{x}}^{rK} \geq 1$, then

$$
\begin{aligned}
\mathbf{x}_2^{(r+1)K-\frac{1}{2}} &= \bar{\mathbf{x}}^{rK} - \eta \sum_{t=rK}^{(r+1)K-1} \nabla f_2(\mathbf{x}_2^t) \\
&= \bar{\mathbf{x}}^{rK} + \eta \sum_{t=rK}^{(r+1)K-1} \frac{L}{2} \mathbf{x}_2^t \\
&\geq 1 + \frac{2}{LK} \sum_{t=rK}^{(r+1)K-1} \frac{L}{2} = 2.
\end{aligned}
$$

Since $\mathbf{x}_1^{(r+1)K-1/2} \geq 0$, we have

$$
\bar{\mathbf{x}}^{(r+1)K} = \frac{1}{2}(\mathbf{x}_1^{(r+1)K-1/2} + \mathbf{x}_2^{(r+1)K-1/2}) \geq 1,
$$

which proves the claim. Therefore, $\mathbf{x}_2^t \geq 1$, $\forall t \in [T]$ and $\mathbf{x}_1^t \geq 0$, $\forall t \in [T]$, which implies $\bar{\mathbf{x}}^t \geq 1/2, \forall t \in [T]$. This shows that $\|\nabla f(\bar{\mathbf{x}}^t)\|^2 \geq L^2/16$, as desired. □

**Remark 1.** *According to Proposition 2, in the worst case a stepsize of $\eta \leq \mathcal{O}(1/K)$ for Local SGD is inevitable. This in view of Vaswani et al. (2019) implies a convergence rate of at most $\mathcal{O}(K/T)$.*

## 4  Numerical Analysis

In this section, we conduct some numerical experiments where we use Local SGD to train an over-parameterized ResNet18 neural network (He et al., 2016) on the Cifar10 dataset (Krizhevsky et al., 2009). This is a standard setting of nonconvex functions under over-parameterization. We also conduct another set of experiments focusing on general convex objective functions, where a perceptron is trained for a synthetic linearly separable binary classification dataset.

### 4.1  ResNet18 Neural Network for Cifar10

We distribute the Cifar10 dataset (Krizhevsky et al., 2009) to $n = 20$ nodes and apply Local SGD to train a ResNet18 neural network (He et al., 2016). The neural network has 11 million trainable parameters and, after sufficient training rounds, can achieve close to 0 training loss, thus satisfying the interpolation property.

For this set of experiments, we run the Local SGD algorithm for $R = 20000$ communication rounds with a different number of local steps per communication round $K = 1, 2, 5, 10, 20$ and report the training error of the global model along the process. We do not report the test accuracy of the model, which is related to the generalization of the model and is beyond the scope of this work[7]. Following the work  Hsieh et al. (2020), we also use Layer Normalization (Ba et al., 2016) instead of Batch Normalization in the architecture of ResNet18 while keeping everything else the same.

We first sort the data by their label, then divide the dataset into 20 shards and assign each of 20 nodes 1 shard. In this way, ten nodes will have image examples of one label, and ten nodes will have image examples of two labels. This regime leads to highly heterogeneous datasets among nodes. We use a training batch size of 8 and choose stepsize $\eta$ to be 0.1 based on a grid search of resolution $10^{-2}$. The simulation results

---

[7]Without data augmentation, the final test accuracy of the model in this set of experiments is around 80%.

are averaged over 3 independent runs of the experiments. We show the global landscape of the result in figures 1a and 1b, where the training loss and the reciprocal of the training loss over the communication rounds are reported, respectively. The decrease in the training loss can be divided into Phase 1, Phase 2, and a transition phase between them, as shown in figures 3a, 3b, and 3c.

**Phase 1:** As figure 3a shows, in the first $\approx 3000$ communication rounds, the reciprocal of the training loss grows nearly linearly with respect to the number of communication rounds. This is strong evidence of the $\mathcal{O}(1/R) = \mathcal{O}(K/T)$ convergence rate of Local SGD as we stated in Theorem 2. We can also see that in this phase, the decrease of training loss depends only on the number of communication rounds $R$ regardless of the number of local steps $K$, thus validating Theorem 2.

**Phase 2:** After $\approx 6000$ communication rounds, as the training loss further decreases (below 0.01), we can observe from figures 3c and figure 2 a clear linear dependence of the reciprocal of the training loss and the total iterations $T$ (notice in figure 2 all lines share similar slope). This corresponds to a $\mathcal{O}(1/T)$ convergence rate of Local SGD. In fact, we conjecture that in this phase, the model has moved close enough to the neighborhood of a global optimal point, which simultaneously minimizes the loss at every single data point. Therefore, every local step moves the model closer to that global optimal point regardless of at which node it is performed, causing the aggregation step to be no longer meaningful and resulting in the convergence rate of $\mathcal{O}(1/T)$ instead of $\mathcal{O}(1/R)$. Another possible explanation is that in Phase 2, the iterates eventually reach a locally convex region and so resemble the convex regime.

The experimental results provided here have two important implications:

- First, from the upper bound in Theorem 2, lower bound in Proposition 2 and the experimental results in Figures 1,2,3, we can imply that for over-parameterized deep learning models, Local SGD indeed converges at a $\mathcal{O}(\frac{1}{T})$ rate[8], which is a strong characterization of the algorithm's actual convergence rate.

- Second, the phenomenon of the two phases also gives us an important empirical implication that in real implementations of the Local SGD algorithm, it might be better to adjust the number of local steps during each communication interval to enforce more frequent communication at first (as in Phase 1 the $\mathcal{O}(1/R) = \mathcal{O}(K/T)$ convergence rate suggests local steps are more or less useless) and less frequent communication later on when one observes the training has entered Phase 2 (by, e.g., observing training loss $\leq 0.05$).

To conclude, we have performed large scale experiments that reveal the convergence behavior of Local SGD for practical over-parameterized deep learning models. We observe from the experiments that the decrease of the training loss can be divided into Phase 1, Phase 2, and a transition phase between them. The convergence rate of Local SGD in practice can be $\mathcal{O}(K/T)$ (Phase 1), or $\mathcal{O}(1/T)$ (Phase 2), or somewhere in between (transition phase). Experimental results in Phase 1 strongly support our theoretical findings in Theorem 2, while experimental results in Phase 2 partially support it and also raise new interesting questions.

### 4.2 Perceptron for Linearly Separable Dataset

We generate a synthetic binary classification dataset with $N = 10000$ data-points uniformly distributed in a $d = 100$ dimensional cube $[-1, 1]^d$. Then, a hyperplane is randomly generated, and all data points above it are labeled '1' with other data points labeled '−1', after which the two sets of points with different labels are "pulled apart" to add a small gap between them. This ensures the dataset is linearly separable and satisfies the interpolation property. We divide the dataset among $n = 16$ nodes and apply Local SGD to distributedly train a perceptron to minimize the finite-sum squared-hinge loss function:

$$f(w) = \frac{1}{N} \sum_{i=1}^{N} \max(0, 1 - y_i x_i^T w)^2.$$

---

[8]If taken into consideration the factor of $K$, then the rate is between $\mathcal{O}(K/T)$ (Phase 1), and $\mathcal{O}(1/T)$ (Phase 2).

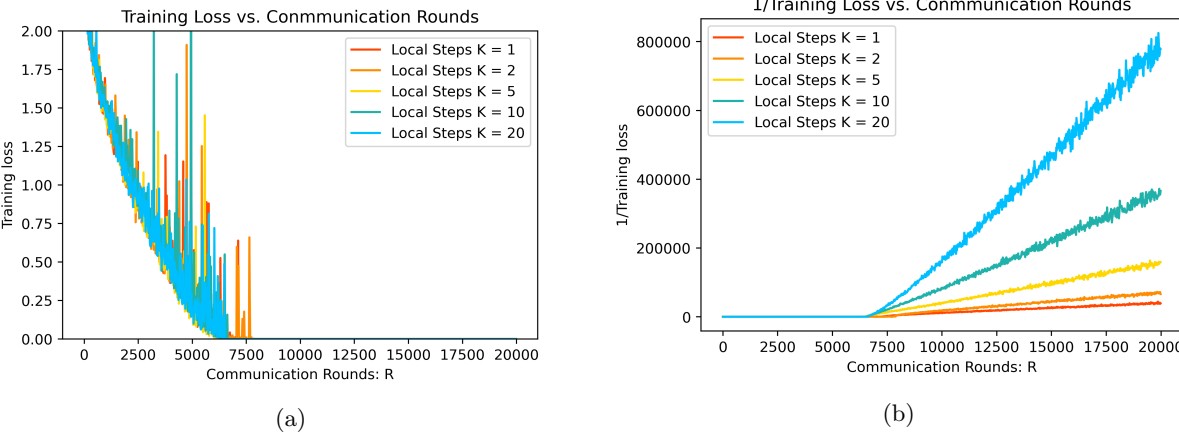

(a)  (b)

Figure 1: 1a: Training loss vs. communication rounds with different local steps. 1b: 1/Training loss vs. communication rounds with different local steps.

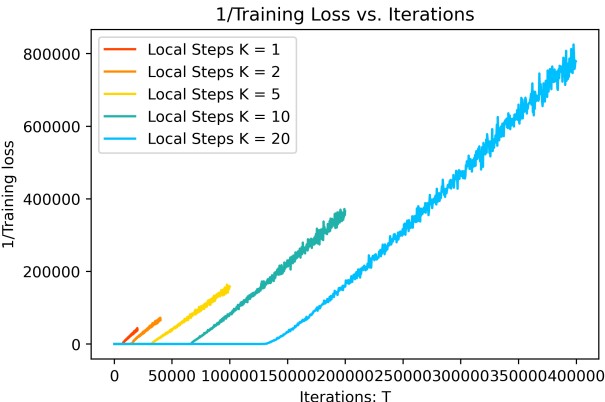

Figure 2: 1/(Training loss) vs. Total number of iterations $T$ with different local steps.

We partition the dataset in three different ways to reflect different data similarity regimes and evaluate the relationship between training loss, communication rounds, and local steps for Local SGD under each of the three regimes. As in Section 4.1, we plot both the training loss and 1/(training loss) vs. the number of communication rounds. We stop the algorithm after at most $10^6$ communication rounds or if the training loss is below $10^{-4}$. We choose stepsize $\eta = 0.075$.

1. **Even partition:** The dataset is partitioned evenly to all nodes, resulting in i.i.d. local data distribution. The simulation results for this regime are shown in Figure 5a,4a.

2. **Pathological partition:** The dataset is partitioned by 17 hyperplanes that are parallel to the initial hyperplane. Distances between adjacent hyperplanes are the same. Each node gets assigned one of the 16 'slices' of data points. This is a highly heterogeneous data partition since 15 out of the 16 nodes will have only one label. The simulation results for this regime are shown in Figure 5b,4b.

3. **Worst case partition:** All data points are assigned to one node. The other 15 nodes have an empty dataset. This partition corresponds to the setting in Example 1. The simulation results for this regime are shown in Figure 5c,4c.

**An $\mathcal{O}(1/T)$ Convergence Rate:** In general, we can clearly observe that the reciprocal of the training loss grows linearly with respect to the number of communication rounds, as well as a linear speedup of the

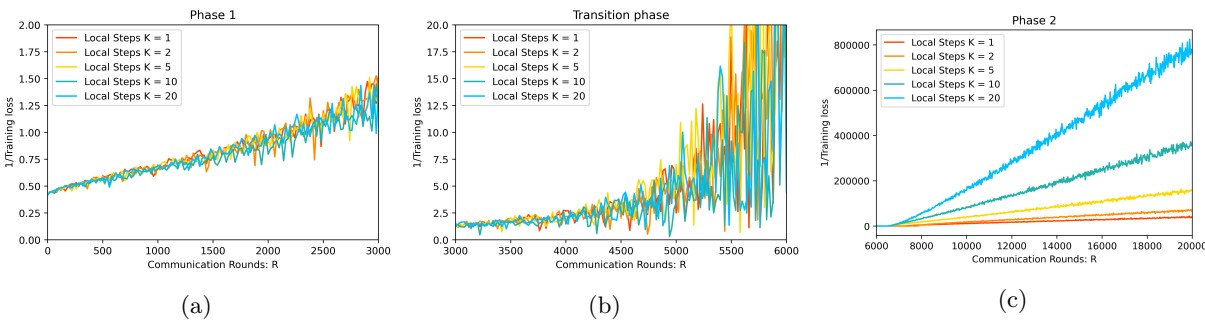

Figure 3: 1/(Training loss) vs. communication rounds for different phases. 3a: Phase 1. 3b: Transition phase. 3c: Phase 2.

convergence rate with the number of local steps in all three regimes. This implies an $\mathcal{O}(\frac{1}{KR}) = \mathcal{O}(1/T)$ convergence rate for Local SGD, validating our result in Theorem 1. The simulation results suggest that despite the $\mathcal{O}(K/T)$ worst-case upper bound, the optimistic $\mathcal{O}(1/T)$ convergence rate in Theorem 1, as well as the effectiveness of local steps, can generally be expected in practice. We also notice cases where Local SGD converges faster than the $\mathcal{O}(1/T)$, we remark that this is also reasonable since Theorem 1 only provides a lower bound for the convergence rate of Loca SGD.

**Effect of Data Heterogeneity:** While in general, Local SGD enjoys an $\mathcal{O}(\frac{1}{KR}) = \mathcal{O}(1/T)$ convergence rate, data heterogeneity is still a key issue and will cause the algorithm to become slower. Comparing the convergence rate of Local SGD under the three different partition regimes, especially the slow convergence of the worst case regime stands in contrast with the similar fast convergence of the other two regimes, we can see that having at least a little data similarity among different nodes is crucial for the convergence rate of the algorithm, as we predicted in Theorem 1.

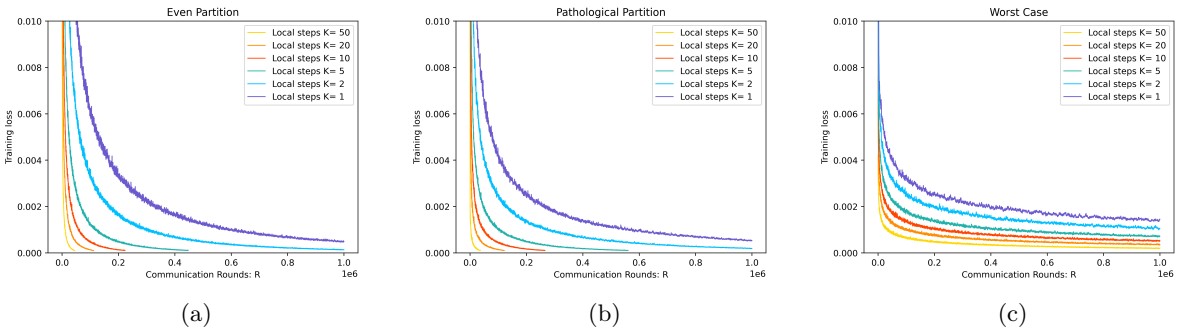

Figure 4: Training loss vs. communication rounds with different local steps under the three data partition regimes. 4a: even partition regime. 4b: pathological partition regime. 4c: worst case partition regime.

## 5 Conclusion

We studied the theoretical convergence guarantees of Local SGD for training over-parameterized models in the heterogeneous data setting and established tight convergence rates for strongly-convex, convex and non-convex loss functions. Moreover, we validated the effectiveness of local steps in speeding up the convergence of Local SGD in various settings both theoretically and using extensive simulations. However, our theoretical results fall short of explaining the effectiveness of local steps in the later phase of training non-convex over-parameterized Neural Networks, as observed in our experiments. We leave this important issue as a future research direction. Our results partially explain the fast convergence of Local SGD (especially when compared to Minibatch SGD) when training large-scale deep learning models.

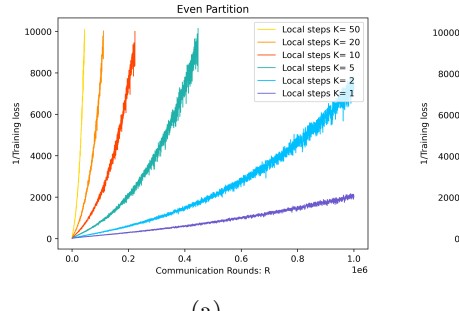 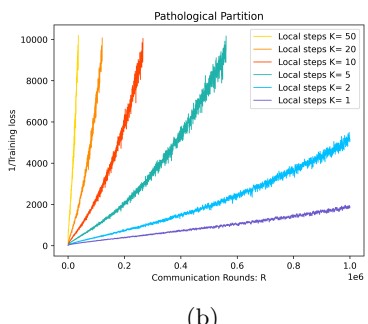 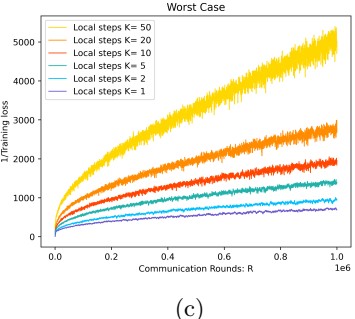

(a)             (b)             (c)

Figure 5: 1/(Training loss) vs. communication rounds with different local steps under the three data partition regimes. 5a: even partition regime. 5b: pathological partition regime. 5c: worst case partition regime.

As future work, one interesting direction would be to generalize our results to the partial node participation setting, which is practical in federated learning. Another interesting direction would be to further study and quantify the two-phase convergence phenomenon of Local SGD when training large-scale neural networks, as we discussed in Section 4.1. This may need combining the interpolation assumption with the special architectures of neural networks (see, e.g., (Allen-Zhu et al., 2019; Du et al., 2019)).

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

# A  Proof of Theorems

Define $\bar{\mathbf{g}}^{(t)} := \frac{1}{n}\sum_{i=1}^{n}\nabla f_i(\mathbf{x}_i^{(t)},\xi_i^{(t)})$ as the average of the stochastic gradients evaluated at all nodes, and $r_t := \mathbb{E}\|\bar{\mathbf{x}}^{(t)} - \mathbf{x}^*\|^2$ as the expected distance to the optimum solution.

## A.1  Preliminary Propositions

**Proposition 3.** *Let* $f : \mathbb{R}^d \to \mathbb{R}$ *be an* $L$-*smooth and convex function and* $\mathbf{x}^* \in \arg\min_{x\in\mathbb{R}^d} f(\mathbf{x})$. *Then,*

$$\frac{1}{2L}\|\nabla f(\mathbf{x})\|^2 \leq f(\mathbf{x}) - f(\mathbf{x}^*) \tag{8}$$

**Proposition 4.** *Let* $\bar{\mathbf{x}} = \frac{1}{n}\sum_{i=1}^{n}\mathbf{x}_i$. *For any* $\mathbf{x}' \in \mathbb{R}^d$, *we have*

$$\sum_{i=1}^{n}\|\mathbf{x}_i - \mathbf{x}'\|^2 = \sum_{i=1}^{n}\|\mathbf{x}_i - \bar{\mathbf{x}}\|^2 + n\|\bar{\mathbf{x}} - \mathbf{x}'\|^2. \tag{9}$$

*As a consequence, we have the following inequalities:*

$$\sum_{i=1}^{n}\|\mathbf{x}_i - \bar{\mathbf{x}}\|^2 \leq \sum_{i=1}^{n}\|\mathbf{x}_i\|^2, \tag{10}$$

$$\|\bar{\mathbf{x}} - \mathbf{x}'\|^2 \leq \frac{1}{n}\sum_{i=1}^{n}\|\mathbf{x}_i - \mathbf{x}'\|^2. \tag{11}$$

## A.2  Proof of Theorem 1

Let $\mathbf{x}^* \in \arg\min_{x\in\mathbb{R}^d} f(\mathbf{x})$. From Assumptions 3 (Interpolation) and 2, we have $\nabla f_i(\mathbf{x}^\star,\xi_i) = 0$, which implies

$$\mathbf{x}^* \in \arg\min_{x\in\mathbb{R}^d} f_i(\mathbf{x},\xi_i),\ \forall i \in [n],\ \xi_i \in \Omega_i$$

.

We first bound the progress made by local variables $\mathbf{x}_i$ in one local SGD update as follows:

**Lemma 2.** *Let Assumption 1, Assumption 2 and Assumption 3 (Interpolation) hold with* $\mu = 0$. *If we follow Algorithm 1 with stepsize* $\eta \leq \frac{1}{2L}$, *we will have*

$$\mathbb{E}_{\xi_i^t}\|\mathbf{x}_i^{t+\frac{1}{2}} - \mathbf{x}^*\|^2 \leq \|\mathbf{x}_i^t - \mathbf{x}^*\|^2 - \eta(f_i(\mathbf{x}_i^t) - f_i(\mathbf{x}^*)) \tag{12}$$

*Proof.*

$$\begin{aligned}
&\mathbb{E}_{\xi_i^t}\|\mathbf{x}_i^{t+\frac{1}{2}} - \mathbf{x}^*\|^2 \\
&= \mathbb{E}_{\xi_i^t}\|\mathbf{x}_i^t - \mathbf{x}^* - \eta\nabla f_i(\mathbf{x}_i^t,\xi_i^t)\|^2 \\
&= \|\mathbf{x}_i^t - \mathbf{x}^*\|^2 - 2\eta\langle\mathbf{x}_i^t - \mathbf{x}^*, \nabla f_i(\mathbf{x}_i^t)\rangle + \eta^2\mathbb{E}_{\xi_i^t}\|\nabla f_i(\mathbf{x}_i^t,\xi_i^t)\|^2 \\
&\overset{(2)(8)}{\leq} \|\mathbf{x}_i^t - \mathbf{x}^*\|^2 - 2\eta(f_i(\mathbf{x}_i) - f_i(\mathbf{x}^*) + \frac{\mu}{2}\|\mathbf{x}_i^t - \mathbf{x}^*\|^2) \\
&\quad + \eta^2\mathbb{E}_{\xi_i^t}[2L(f_i(\mathbf{x}_i^t,\xi_i^t) - f_i(\mathbf{x}^*,\xi_i^t))] \\
&= (1 - \eta\mu)\|\mathbf{x}_i^t - \mathbf{x}^*\|^2 - (2\eta - 2L\eta^2)(f_i(\mathbf{x}_i) - f_i(\mathbf{x}^*)) \\
&\overset{(\eta\leq\frac{1}{2L},\mu=0)}{\leq} \|\mathbf{x}_i^t - \mathbf{x}^*\|^2 - \eta(f_i(\mathbf{x}_i^t) - f_i(\mathbf{x}^*)).
\end{aligned}$$

$\square$

Next, we bound the progress made by $\bar{\mathbf{x}}$ in one communication round as follows:

**Lemma 3.** *Let Assumption 1, Assumption 2 and Assumption 3 (Interpolation) hold with $\mu = 0$. Assume that the nodes follow Algorithm 1 with stepsize $\eta \leq \frac{1}{2L}$, and let $w_t = 1$ if $t \in \mathcal{I}$ or $t + 1 \in \mathcal{I}$, and $w_t = c$ otherwise. Then, for $r = 0, 1, \ldots, R - 1$, we have*

$$\mathbb{E}\|\bar{\mathbf{x}}^{(r+1)K} - \mathbf{x}^*\|^2 \leq \mathbb{E}\|\bar{\mathbf{x}}^{rK} - \mathbf{x}^*\|^2 - \eta \sum_{t=rK}^{(r+1)K-1} w_t \mathbb{E}[f(\bar{\mathbf{x}}^t) - f^*].$$

*Proof.*

$$\mathbb{E}\|\bar{\mathbf{x}}^{(r+1)K} - \mathbf{x}^*\|^2 = \mathbb{E}\|\frac{1}{n}\sum_{i=1}^n \mathbf{x}_i^{(r+1)K-\frac{1}{2}} - \mathbf{x}^*\|^2$$

$$\stackrel{(9)}{=} \frac{1}{n}\sum_{i=1}^n \mathbb{E}\|\mathbf{x}_i^{(r+1)K-\frac{1}{2}} - \mathbf{x}^*\|^2 - \frac{1}{n}\sum_{i=1}^n \mathbb{E}\|\mathbf{x}_i^{(r+1)K-\frac{1}{2}} - \bar{\mathbf{x}}^{(r+1)K}\|^2$$

$$\stackrel{(12)}{\leq} \frac{1}{n}\sum_{i=1}^n \mathbb{E}\Big[\|\mathbf{x}_i^{rK} - \mathbf{x}^*\|^2 - \eta \sum_{t=rK}^{(r+1)K-1} (f_i(\mathbf{x}_i^t) - f_i(\mathbf{x}^*))\Big] - \frac{1}{n}\sum_{i=1}^n \mathbb{E}\|\mathbf{x}_i^{(r+1)K-\frac{1}{2}} - \bar{\mathbf{x}}^{(r+1)K}\|^2$$

$$\stackrel{(5)}{\leq} \mathbb{E}\|\bar{\mathbf{x}}^{rK} - \mathbf{x}^*\|^2 - \frac{\eta}{n}\sum_{i=1}^n \mathbb{E}[f_i(\bar{\mathbf{x}}^{rK}) - f_i(\mathbf{x}^*)] - \eta \sum_{i=1}^n \sum_{t=rK+1}^{(r+1)K-2} c\mathbb{E}[f(\bar{\mathbf{x}}^t) - f(\mathbf{x}^*))]$$

$$- \frac{\eta}{n}\sum_{i=1}^n \mathbb{E}[f_i(\mathbf{x}_i^{(r+1)K-1}) - f_i(\mathbf{x}^*)] - \frac{1}{n}\sum_{i=1}^n \mathbb{E}\|\mathbf{x}_i^{(r+1)K-\frac{1}{2}} - \bar{\mathbf{x}}^{(r+1)K}\|^2$$

$$= \mathbb{E}\|\bar{\mathbf{x}}^{rK} - \mathbf{x}^*\|^2 - \eta \sum_{t=rK}^{(r+1)K-2} w_t \mathbb{E}[f(\bar{\mathbf{x}}^t) - f^*] - \frac{\eta}{n}\sum_{i=1}^n \mathbb{E}[f_i(\mathbf{x}_i^{(r+1)K-1}) - f_i(\mathbf{x}^*)]$$

$$\underbrace{- \frac{1}{n}\sum_{i=1}^n \mathbb{E}\|\mathbf{x}_i^{(r+1)K-\frac{1}{2}} - \bar{\mathbf{x}}^{(r+1)K}\|^2}_{T_1}.$$

Since $T_1 \geq 0$, we can bound $T_1$ as

$$T_1 = \frac{1}{n}\sum_{i=1}^n \mathbb{E}\|\mathbf{x}_i^{(r+1)K-1} - \bar{\mathbf{x}}^{(r+1)K-1} - \eta\nabla f_i(\mathbf{x}_i^{(r+1)K-1}, \xi_i^{(r+1)K-1}) + \eta\bar{\mathbf{g}}^{(r+1)K-1}\|^2$$

$$= \frac{1}{n}\sum_{i=1}^n \mathbb{E}\|\mathbf{x}_i^{(r+1)K-1} - \bar{\mathbf{x}}^{(r+1)K-1}\|^2 + \eta^2 \frac{1}{n}\sum_{i=1}^n \mathbb{E}\|f_i(\mathbf{x}_i^{(r+1)K-1}, \xi_i^{(r+1)K-1}) + \bar{\mathbf{g}}^{(r+1)K-1}\|^2$$

$$- 2\eta\frac{1}{n}\sum_{i=1}^n \mathbb{E}[\langle \mathbf{x}_i^{(r+1)K-1} - \bar{\mathbf{x}}^{(r+1)K-1}, \nabla f_i(\mathbf{x}_i^{(r+1)K-1}) - \bar{\mathbf{g}}^{(r+1)K-1}\rangle]$$

$$\geq \frac{1}{n}\sum_{i=1}^n \mathbb{E}\|\mathbf{x}_i^{(r+1)K-1} - \bar{\mathbf{x}}^{(r+1)K-1}\|^2 - 2\eta\frac{1}{n}\sum_{i=1}^n \mathbb{E}\Big[f_i(\bar{\mathbf{x}}^{(r+1)K-1}) - f_i(\mathbf{x}_i^{(r+1)K-1})$$

$$+ \frac{L}{2}\|\mathbf{x}_i^{(r+1)K-1} - \bar{\mathbf{x}}^{(r+1)K-1}\|^2\Big]$$

$$\stackrel{(\eta \leq \frac{1}{2L})}{\geq} 2\eta\frac{1}{n}\sum_{i=1}^n \mathbb{E}[f_i(\bar{\mathbf{x}}^{(r+1)K-1}) - f_i(\mathbf{x}_i^{(r+1)K-1})]$$

$$= 2\eta\mathbb{E}[f(\bar{\mathbf{x}}^{(r+1)K-1})] - \frac{2\eta}{n}\sum_{i=1}^n \mathbb{E}[f_i(\mathbf{x}_i^{(r+1)K-1})].$$

Therefore,

$$T_1 \geq \frac{T_1}{2} \geq \eta \mathbb{E}[f(\bar{\mathbf{x}}^{(r+1)K-1})] - \frac{\eta}{n} \sum_{i=1}^n \mathbb{E}[f_i(\mathbf{x}_i^{(r+1)K-1})].$$

Substituting back we get

$$\mathbb{E}\|\bar{\mathbf{x}}^{(r+1)K} - \mathbf{x}^*\|^2$$

$$\leq \mathbb{E}\|\bar{\mathbf{x}}^{rK} - \mathbf{x}^*\|^2 - \eta \sum_{t=rK}^{(r+1)K-2} w_t \mathbb{E}[f(\bar{\mathbf{x}}^t) - f^*] - \frac{\eta}{n} \sum_{i=1}^n \mathbb{E}[f_i(\mathbf{x}_i^{(r+1)K-1}) - f_i(\mathbf{x}^*)]$$

$$- \left(\eta \mathbb{E}[f(\bar{\mathbf{x}}^{(r+1)K-1})] - \frac{\eta}{n} \sum_{i=1}^n \mathbb{E}[f_i(\mathbf{x}_i^{(r+1)K-1})]\right)$$

$$= \mathbb{E}\|\bar{\mathbf{x}}^{rK} - \mathbf{x}^*\|^2 - \eta \sum_{t=rK}^{(r+1)K-2} w_t \mathbb{E}[f(\bar{\mathbf{x}}^t) - f^*] - \eta \mathbb{E}[f(\bar{\mathbf{x}}^{(r+1)K-1}) - f(\mathbf{x}^*)]$$

$$= \mathbb{E}\|\bar{\mathbf{x}}^{rK} - \mathbf{x}^*\|^2 - \eta \sum_{t=rK}^{(r+1)K-1} w_t \mathbb{E}[f(\bar{\mathbf{x}}^t) - f^*].$$

$\square$

To complete the proof of Theorem 1, using Lemma 3, we can write

$$\mathbb{E}\|\bar{\mathbf{x}}^{(T)} - \mathbf{x}^*\|^2 = \mathbb{E}\|\bar{\mathbf{x}}^{RK} - \mathbf{x}^*\|^2$$

$$\leq \mathbb{E}\|\bar{\mathbf{x}}^{(R-1)K} - \mathbf{x}^*\|^2 - \eta \sum_{t=(R-1)K}^{RK-1} w_t \mathbb{E}[f(\bar{\mathbf{x}}^t) - f^*]$$

$$\leq \cdots \leq \|\mathbf{x}^{(0)} - \mathbf{x}^*\|^2 - \eta \sum_{t=0}^{T-1} w_t \mathbb{E}[f(\bar{\mathbf{x}}^t) - f^*].$$

Therefore, we can write

$$\frac{1}{W} \sum_{t=0}^{T-1} w_t \mathbb{E}[f(\bar{\mathbf{x}}^t) - f^*] \leq \frac{\|\mathbf{x}^{(0)} - \mathbf{x}^*\|^2}{\eta W} = \frac{K\|\mathbf{x}^{(0)} - \mathbf{x}^*\|^2}{\eta(cKT + 2(1-c)T)}.$$

Theorem 1 now follows from Jensen's inequality.

### A.3   Proof of Theorem 2

Define $V_t := \frac{1}{n} \mathbb{E} \sum_{i=1}^n \|\mathbf{x}_i^{(t)} - \bar{\mathbf{x}}^{(t)}\|^2$ to be the expected consensus error and $e_t := \mathbb{E}f(\bar{\mathbf{x}}^{(t)}) - f(\mathbf{x}^*)$ to be the expected optimality gap. Moreover, let $h_t := \mathbb{E}\|\nabla f(\bar{\mathbf{x}}^{(t)})\|^2$ be the expected gradient norm of the average iterate.

We first establish the following descent lemma to bound the progress of $\bar{\mathbf{x}}^t$ in one iteration:

**Lemma 4.** *Let Assumption 1, Assumption 4 (SGC) hold. If we follow Algorithm 1 with stepsize $\eta \leq \frac{1}{3KL\rho}$ and $K \geq 2$, we have*

$$e_{t+1} \leq e_t - \frac{1}{3}\eta h_t + \frac{2}{3}\eta L^2 V_t \tag{13}$$

*Proof.*

$$\mathbb{E}f(\bar{\mathbf{x}}^{(t+1)}) = \mathbb{E}f(\bar{\mathbf{x}}^{(t)} - \eta\frac{1}{n}\sum_{i=1}^{n}\nabla f_i(\mathbf{x}_i^{(t)}, \xi_i^{(t)})$$

$$\leq \mathbb{E}f(\bar{\mathbf{x}}^{(t)}) \underbrace{- \eta\mathbb{E}\langle\nabla f(\bar{\mathbf{x}}^{(t)}), \frac{1}{n}\sum_{i=1}^{n}\nabla f_i(\mathbf{x}_i^{(t)}, \xi_i^{(t)})\rangle}_{T_1} + \underbrace{\frac{L}{2}\eta^2\mathbb{E}\|\frac{1}{n}\sum_{i=1}^{n}\nabla f_i(\mathbf{x}_i^{(t)}, \xi_i^{(t)})\|^2}_{T_2}.$$

To bound the term $T_1$, we can write

$$\mathbb{E}\langle\nabla f(\bar{\mathbf{x}}^{(t)}), \frac{1}{n}\sum_{i=1}^{n}\nabla f_i(\mathbf{x}_i^{(t)}, \xi_i^{(t)})\rangle$$

$$=\mathbb{E}\langle\nabla f(\bar{\mathbf{x}}^{(t)}), \frac{1}{n}\sum_{i=1}^{n}\nabla f_i(\mathbf{x}_i^{(t)})\rangle$$

$$=\mathbb{E}\|\nabla f(\bar{\mathbf{x}}^{(t)})\|^2 + \mathbb{E}\langle\nabla f(\bar{\mathbf{x}}^{(t)}), \frac{1}{n}\sum_{i=1}^{n}(\nabla f_i(\mathbf{x}_i^{(t)}) - \nabla f_i(\bar{\mathbf{x}}^{(t)}))\rangle$$

$$\geq\frac{1}{2}\mathbb{E}\|\nabla f(\bar{\mathbf{x}}^{(t)})\|^2 - \frac{1}{2n}\sum_{i=1}^{n}\mathbb{E}\|\nabla f_i(\mathbf{x}_i^{(t)}) - \nabla f_i(\bar{\mathbf{x}}^{(t)})\|^2$$

$$\geq\frac{1}{2}\mathbb{E}\|\nabla f(\bar{\mathbf{x}}^{(t)})\|^2 - \frac{L^2}{2n}\sum_{i=1}^{n}\mathbb{E}\|\bar{\mathbf{x}}^{(t)} - \mathbf{x}_i^{(t)}\|^2$$

$$=\frac{1}{2}h_t - \frac{L^2}{2}V_t,$$

where in the third inequality we have used $\langle a, b\rangle \geq -\frac{1}{2}\|a\|^2 - \frac{1}{2}\|b\|^2$. Next, in order to bound $T_2$, we have

$$\mathbb{E}\|\frac{1}{n}\sum_{i=1}^{n}\nabla f_i(\mathbf{x}_i^t, \xi_i^t)\|^2 \leq \frac{1}{n}\mathbb{E}\sum_{i=1}^{n}\|\nabla f_i(\mathbf{x}_i^t, \xi_i^t)\|^2$$

$$\overset{(3)}{\leq}\frac{\rho}{n}\mathbb{E}\sum_{i=1}^{n}\|\nabla f(\mathbf{x}_i^t)\|^2$$

$$\leq\frac{2\rho}{n}\mathbb{E}\sum_{i=1}^{n}\|\nabla f(\bar{\mathbf{x}}^t)\|^2 + \frac{2\rho}{n}\mathbb{E}\sum_{i=1}^{n}\|\nabla f(\mathbf{x}_i^t) - \nabla f(\bar{\mathbf{x}}^t)\|^2$$

$$\leq 2\rho h_t + 2L^2\rho V_t.$$

Putting everything together and subtracting $f(\mathbf{x}^*)$ from both sides of the resulting inequality, we get

$$e_{t+1} \leq e_t - \eta(\frac{1}{2}h_t - \frac{L^2}{2}V_t) + \frac{L\eta^2}{2}(2\rho h_t + 2L^2\rho V_t)$$

$$\overset{(\eta\leq\frac{1}{6L\rho})}{\leq} e_t - \frac{1}{3}\eta h_t + \frac{2}{3}\eta L^2 V_t.$$

$\square$

Next, we bound the consensus error $V_t$ using the following lemma:

**Lemma 5.** *Let Assumption 1, Assumption 4 (SGC) hold. Assume the nodes follow Algorithm 1 with stepsize $\eta \leq \frac{1}{3KL\rho}$, and define $\tau(t) \coloneqq \max_{s\leq t, s\in\mathcal{I}} s$. Then,*

$$V_t \leq 3\eta^2 K\rho \sum_{j=\tau(t)}^{t-1} h_j. \tag{14}$$

*Proof.*

$$nV_t = \mathbb{E}\sum_{i=1}^{n}\|\mathbf{x}_i^t - \bar{\mathbf{x}}^t\|^2$$

$$= \sum_{i=1}^{n}\mathbb{E}\|(\mathbf{x}_i^{\tau(t)} - \sum_{j=\tau(t)}^{t-1}\eta\nabla f_i(\mathbf{x}_i^j,\xi_i^j)) - (\bar{\mathbf{x}}^{\tau(t)} - \sum_{j=\tau(t)}^{t-1}\eta\bar{\mathbf{g}}^j)\|^2$$

$$= \sum_{i=1}^{n}\mathbb{E}\| - \sum_{j=\tau(t)}^{t-1}\eta\nabla f_i(\mathbf{x}_i^j,\xi_i^j) + \sum_{j=\tau(t)}^{t-1}\eta\bar{\mathbf{g}}^j\|^2$$

$$\overset{(10)}{\leq} \sum_{i=1}^{n}\mathbb{E}\|\sum_{j=\tau(t)}^{t-1}\eta\nabla f_i(\mathbf{x}_i^j,\xi_i^j)\|^2$$

$$\leq \eta^2(t-\tau(t))\sum_{i=1}^{n}\sum_{j=\tau(t)}^{t-1}\mathbb{E}\|\nabla f_i(\mathbf{x}_i^j,\xi_i^j)\|^2$$

$$\overset{(3)}{\leq} \eta^2(t-\tau(t))\rho\sum_{i=1}^{n}\sum_{j=\tau(t)}^{t-1}\mathbb{E}\|\nabla f(\mathbf{x}_i^j)\|^2$$

$$\leq 2\eta^2(t-\tau(t))\rho\sum_{i=1}^{n}\sum_{j=\tau(t)}^{t-1}\mathbb{E}\Big[\|\nabla f(\bar{\mathbf{x}}^j) - \nabla f(\mathbf{x}_i^j)\|^2 + \|\nabla f(\bar{\mathbf{x}}^j)\|^2\Big]$$

$$\leq 2n\eta^2 K\rho\sum_{j=\tau(t)}^{t-1}h_j + 2n\eta^2 K\rho L^2\sum_{j=\tau(t)}^{t-1}V_j.$$

Since $\eta \leq \frac{1}{3KL\rho}$, we have

$$V_t \leq 2\eta^2 K\rho\sum_{j=\tau(t)}^{t-1}h_j + \frac{1}{4K\rho}\sum_{j=\tau(t)}^{t-1}V_j.$$

Unrolling all $V_j, j = \tau(t),\ldots,t-1$, and noting that $\rho \geq 1$, we have

$$V_t \leq \frac{1}{4K\rho}\sum_{j=\tau(t)}^{t-1}V_j + 2\eta^2 K\rho\sum_{j=\tau(t)}^{t-1}h_j$$

$$\leq \frac{1}{4K\rho}\sum_{j=\tau(t)}^{t-2}V_j 2\eta^2 K\rho\sum_{j=\tau(t)}^{t-1}h_j + \frac{1}{4K\rho}(\frac{1}{4K\rho}\sum_{j=\tau(t)}^{t-2}V_j + 2\eta^2 K\rho\sum_{j=\tau(t)}^{t-2}h_j)$$

$$\leq \cdots \leq (1+\frac{1}{4K\rho})^K 2\eta^2 K\rho\sum_{j=\tau(t)}^{t-1}h_j$$

$$\leq 3\eta^2 K\rho\sum_{j=\tau(t)}^{t-1}h_j.$$

$\square$

To complete the proof of Theorem 2, we combine (13) and (14) by applying a telescoping sum on (13) to get

$$
\frac{1}{3}\eta \sum_{t=0}^{T-1} h_t \leq e_0 + \frac{2}{3}\eta L^2 \sum_{t=0}^{T-1} V_t
$$

$$
\stackrel{(14)}{\leq} e_0 + 2\eta L^2 \sum_{t=0}^{T-1} \eta^2 K \rho \sum_{j=\tau(t)}^{t-1} h_j
$$

$$
= e_0 + 2\eta^3 L^2 K \rho \sum_{j=0}^{T-2} h_j \sum_{t=j+1}^{\tau(j)+K} 1
$$

$$
\leq e_0 + 2\eta^3 L^2 K^2 \rho \sum_{t=0}^{T-2} h_t
$$

$$
\stackrel{(\eta \leq \frac{1}{3KL\rho})}{\leq} e_0 + \frac{2}{9\rho}\eta \sum_{t=0}^{T-1} h_t
$$

$$
\leq e_0 + \frac{2}{9}\eta \sum_{t=0}^{T-1} h_t.
$$

Therefore, we have

$$
\frac{1}{T}\sum_{t=0}^{T-1} h_t \leq \frac{9e_0}{\eta T} \;\; \Rightarrow \;\; \min_{0 \leq t \leq T-1} h_t \leq \frac{9e_0}{\eta T}.
$$

This completes the proof for Theorem 2.

## B  Experimental Verification of Assumption 4 (SGC)

Here, we perform experimental verification of Assumption 4 (SGC) with the same problem setup as in Section 4.1. For verification purposes, we use centralized SGD to train the aforementioned ResNet18 neural network on the Cifar10 dataset for a total of 1000 epochs, and we plot the global gradient norm and the maximum of per-sample gradient norm of the model as well as the ratio between the two gradient norms in Figure 6.

It is shown in Figure 6 that throughout the training, the ratio $r = \frac{\max_{i,\xi_i} \|\nabla f_i(\mathbf{x}, \xi_i)\|}{\|\nabla f(\mathbf{x})\|}$ never exceeds 4000, and after a transient phase quickly stablizes aroung 500. Since Assumption 4 requires that

$$
\mathbb{E}_{\xi_i \sim \mathcal{D}_i} \|\nabla f_i(\mathbf{x}, \xi_i)\|^2 \leq \rho \|\nabla f(\mathbf{x})\|^2, \; \forall i,
$$

while we always have

$$
\mathbb{E}_{\xi_i \sim \mathcal{D}_i} \|\nabla f_i(\mathbf{x}, \xi_i)\|^2 \leq \max_{\xi_i} \|\nabla f_i(\mathbf{x}, \xi_i)\|^2
$$

and in most cases

$$
\mathbb{E}_{\xi_i \sim \mathcal{D}_i} \|\nabla f_i(\mathbf{x}, \xi_i)\|^2 \ll \max_{\xi_i} \|\nabla f_i(\mathbf{x}, \xi_i)\|^2,
$$

we have verified that Assumption 4 (SGC) is a valid assumption for over-parameterized models in practice.

## C  $\mathcal{O}(\exp(-T))$ Convergence for Strongly Convex Loss Functions

For strongly convex loss functions, an error bound of $\mathcal{O}(\exp(-T))$ can be achieved under the over-parameterized setting, where $T$ is the total number of iterations. Before our work, the best-known convergence rate was $\mathcal{O}(\exp(-T/K))$ (Qu et al., 2020; Koloskova et al., 2020).

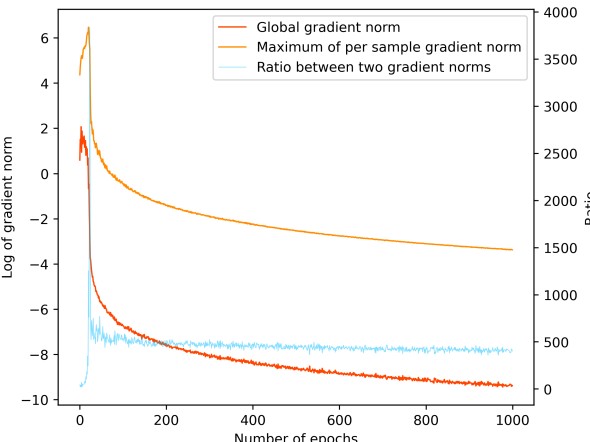

Figure 6: The global gradient norm, maximum of per-sample gradient norm and the ratio between them vs. number of epochs. The red line is the global gradient norm and the orange line is the maximum of per-sample gradient norm, they are plotted in log scale whose values correspond tothe y axis on the left. The blue line is the ratio between the two norms whose value correspond tothe y axis on the right.

**Theorem 3** (Strongly convex functions)**.** *Let Assumption 1, Assumption 2 and Assumption 3 (Interpolation) hold with $\mu > 0$. If we follow Algorithm 1 with stepsize $\eta \leq 1/L$, we will have*

$$\mathbb{E}\|\bar{\mathbf{x}}^{(T)} - \mathbf{x}^*\|^2 \leq \left(1 - \eta\mu\right)^T \|\mathbf{x}^{(0)} - \mathbf{x}^*\|^2,$$

*where $\bar{\mathbf{x}}^{(t)} := \frac{1}{n}\sum_{i=1}^{n}\mathbf{x}_i^{(t)}$ is the average of all nodes' iterates at time step $t$. As a special case, if we choose $\eta = 1/L$, then*

$$\mathbb{E}\|\bar{\mathbf{x}}^{(T)} - \mathbf{x}^*\|^2 \leq \left(1 - \frac{\mu}{L}\right)^T \|\mathbf{x}^{(0)} - \mathbf{x}^*\|^2. \tag{15}$$

It was shown in Qu et al. (2020); Koloskova et al. (2020) that Local SGD achieves a geometric convergence rate for strongly convex loss functions in the over-parameterized setting. However, both Qu et al. (2020) and Koloskova et al. (2020) give an $\mathcal{O}(\exp(-T/K))$ convergence rate, while our convergence rate is $\mathcal{O}(\exp(-T))$. The difference between these two rates is significant because the former rate implies that local steps do not contribute to the error bound (since the convergence rate essentially depends on the number of communication rounds $R = T/K$). In contrast, the latter rate suggests local steps can drive the iterates to the optimal solution exponentially fast. The difference between the rates in Qu et al. (2020); Koloskova et al. (2020) and Theorem 3 can be explained by the fact that Qu et al. (2020); Koloskova et al. (2020) use a smaller stepsize of $\eta = \mathcal{O}(\frac{1}{KL})$, while our analysis allows a larger stepsize of $\eta = 1/L$.

### C.1 Proof of Theorem 3

Let $\mathbf{x}^* \in \arg\min_{x \in \mathbb{R}^d} f(\mathbf{x})$. From Assumptions 3 (Interpolation) and 2, we have $\nabla f_i(\mathbf{x}^\star, \xi_i) = 0$, which implies $\mathbf{x}^* \in \arg\min_{x \in \mathbb{R}^d} f_i(\mathbf{x}, \xi_i), \ \forall i \in [n], \ \xi_i \in \Omega_i$.

We first bound the progress made by local variables $\mathbf{x}_i$ in one local SGD update as follows:

**Lemma 6.** *Let Assumption 1, Assumption 2 and Assumption 3 (Interpolation) hold with $\mu > 0$. If we follow Algorithm 1 with stepsize $\eta \leq \frac{1}{L}$, we will have*

$$\mathbb{E}_{\xi_i^t}\|\mathbf{x}_i^{t+\frac{1}{2}} - \mathbf{x}^*\|^2 \leq (1 - \eta\mu)\|\mathbf{x}_i^t - \mathbf{x}^*\|^2 \tag{16}$$

*Proof.*

$$\mathbb{E}_{\xi_i^t} \|\mathbf{x}_i^{t+\frac{1}{2}} - \mathbf{x}^*\|^2$$
$$= \mathbb{E}_{\xi_i^t} \|\mathbf{x}_i^t - \mathbf{x}^* - \eta \nabla f_i(\mathbf{x}_i^t, \xi_i^t)\|^2$$
$$= \|\mathbf{x}_i^t - \mathbf{x}^*\|^2 - 2\eta \langle \mathbf{x}_i^t - \mathbf{x}^*, \nabla f_i(\mathbf{x}_i^t) \rangle + \eta^2 \mathbb{E}_{\xi_i^t} \|\nabla f_i(\mathbf{x}_i^t, \xi_i^t)\|^2$$
$$\overset{(2)(8)}{\leq} \|\mathbf{x}_i^t - \mathbf{x}^*\|^2 - 2\eta(f_i(\mathbf{x}_i) - f_i(\mathbf{x}^*) + \frac{\mu}{2}\|\mathbf{x}_i^t - \mathbf{x}^*\|^2)$$
$$+ \eta^2 \mathbb{E}_{\xi_i^t}[2L(f_i(\mathbf{x}_i^t, \xi_i^t) - f_i(\mathbf{x}^*, \xi_i^t))]$$
$$= (1 - \eta\mu)\|\mathbf{x}_i^t - \mathbf{x}^*\|^2 - (2\eta - 2L\eta^2)(f_i(\mathbf{x}_i) - f_i(\mathbf{x}^*))$$
$$\overset{(\eta \leq \frac{1}{L})}{=} (1 - \eta\mu)\|\mathbf{x}_i^t - \mathbf{x}^*\|^2.$$

$\square$

Using Lemma 6 and Proposition 4, we can bound the progress made by $\bar{\mathbf{x}}$ in one communication round as follows:

**Lemma 7.** *Let Assumption 1, Assumption 2 and Assumption 3 (Interpolation) hold with $\mu > 0$. If we follow Algorithm 1 with stepsize $\eta \leq \frac{1}{L}$, we will have*

$$\mathbb{E}\|\bar{\mathbf{x}}^{(r+1)K} - \mathbf{x}^*\|^2 \leq (1 - \eta\mu)^K \mathbb{E}\|\bar{\mathbf{x}}^{rK} - \mathbf{x}^*\|^2,$$

*for $r = 0, 1, \ldots, R - 1$.*

*Proof.*

$$\mathbb{E}\|\bar{\mathbf{x}}^{(r+1)K} - \mathbf{x}^*\|^2 = \mathbb{E}\|\frac{1}{n}\sum_{i=1}^{n} \mathbf{x}_i^{(r+1)K - \frac{1}{2}} - \mathbf{x}^*\|^2$$
$$\overset{(11)}{\leq} \frac{1}{n}\sum_{i=1}^{n} \mathbb{E}\|\mathbf{x}_i^{(r+1)K - \frac{1}{2}} - \mathbf{x}^*\|^2$$
$$\overset{(16)}{\leq} \frac{1}{n}\sum_{i=1}^{n} (1 - \eta\mu)^K \mathbb{E}\|\mathbf{x}_i^{rK} - \mathbf{x}^*\|^2$$
$$= (1 - \eta\mu)^K \mathbb{E}\|\bar{\mathbf{x}}^{rK} - \mathbf{x}^*\|^2.$$

$\square$

The proof of Theorem 3 now follows by simply noting that

$$\mathbb{E}\|\bar{\mathbf{x}}^{(T)} - \mathbf{x}^*\|^2 = \mathbb{E}\|\bar{\mathbf{x}}^{RK} - \mathbf{x}^*\|^2$$
$$\leq (1 - \eta\mu)^K \mathbb{E}\|\bar{\mathbf{x}}^{(R-1)K} - \mathbf{x}^*\|^2$$
$$\leq \cdots \leq (1 - \eta\mu)^T \|\mathbf{x}^{(0)} - \mathbf{x}^*\|^2$$

## C.2 Perceptron for Linearly Separable Dataset, Strongly Convex Case

To evaluate the performance of Local SGD for the over-parameterized model with strongly-convex loss functions, we adopt the same experimental setup as in Section 4.2 but add a correction term to the squared-hinge loss to make it strongly convex and run another experiment on the pathologically partitioned dataset. The result is shown in Figure 7. The $\mathcal{O}(\exp(-KR)) = \mathcal{O}(\exp(-T))$ convergence rate can be observed from the figure, validating our result in Theorem 3.

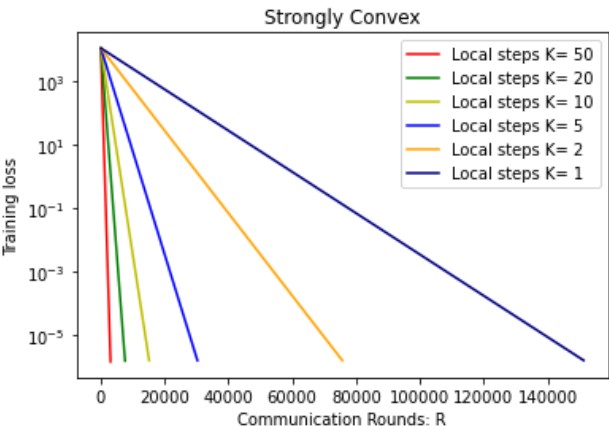

Figure 7: Local SGD for the over-parameterized model with strongly-convex loss functions. Training loss vs. communication rounds with different local steps under the pathological data partition regimes. Training loss is in log scale.

## D   Discussion on the proof in Zhang & Li (2021)

In Section 9.3.2. Discussion on Theorem 3 of the paper Zhang & Li (2021), the authors stated that $M_n \geq \min_i \frac{T_i}{L_i^2}$, which is essential to their result in the Discussion, which can be interpreted as an $\mathcal{O}(\frac{1}{\sqrt{T}})$ convergence rate. However, this inequality does not hold. A simple counterexample is when one of the local nodes finds the optimal point after the first local step, in which case $h_{i,n}(0) = 1$ and $h_{i,n}(t) = 0$ for all $t = 1, 2, \ldots T_i - 1$, and $M_n = \min_i \alpha_i \sum_{t=0}^{T_i-1} h_{i,n}(t) \leq \alpha_i = \frac{1}{L_i^2}$. However, this contradicts the claimed inequality $M_n \geq \min_i \frac{T_i}{L_i^2}$.

