# OpenReview forum: "Faster Convergence of Local SGD for Over-Parameterized Models"
_TMLR — Accepted by TMLR_

### Review · Reviewer_cCmn · 2023-12-25

**Summary Of Contributions:**

This paper explores the area of distributed stochastic optimization, specifically examining the effectiveness of Local SGD. It merges $L$-smoothness and over-parametrization with data similarity conditions, addressing both (strongly-)convex and non-convex scenarios. Theoretical results presented here advance beyond previous studies and are supported with some lower-bound examples. The paper also includes numerical experiments to corroborate its mathematical assertions.

**Audience:**

Yes

**Broader Impact Concerns:**

No for this work

**Claims And Evidence:**

No

**Requested Changes:**

**C1.** The paper should clarify how its analysis differs from prior works, elaborating on the technical novelties and what it is based upon.

**C2.** Definition 1 appears non-standard in the context of distributed learning literature. A more comprehensive discussion on this, particularly if it acts as an additional assumption on local losses $f_i$, would strengthen the paper.

**Strengths And Weaknesses:**

# Strengths

**1.** The manuscript is well-composed, with a clear presentation and delineation of contributions.

**2.** The paper addresses an important and already long-standing problem of understanding a practical distributed training method: Local SGD.

**3.** Analysis is done for various settings: strongly convex, convex, and non-convex loss functions.

**4.** Experiments are performed for both a synthetic problem and a practical task with real data.

# Weaknesses

**W1.** The paper falls short in demonstrating a decisive improvement in performance through the use of local steps, akin to previous studies on Local SGD. The convergence bounds are optimized for the minimum $K$, leaving ambiguity about the additional methodological insights derived from this research. Besides, the discussion after Theorem 1 says that
> local steps can speed up the convergence of Local SGD by at least factor 2

However, the explanation provided is unclear. It does not say how exactly the number of local steps affects convergence.

Theorem 2 doesn’t demonstrate any advantage of using local steps, showing results typical for local methods, which necessitate scaling the step size inversely to the number of local steps. I am curious if this result improves upon previous studies.

However, the conclusion says that
> Moreover, we validated the eﬀectiveness of local steps in **speeding up the convergence** of Local SGD in various settings both theoretically and using extensive simulations.

which seems to be not fully supported.

**W2.** The lower bound results rely on a highly specific set of parameters $n, R$, limiting their general applicability.
It is known that Gradient Descent can provably diverge for step size $\eta > 1/L$ (Nesterov, 2018). Thus, it would be odd to expect a different result for a variation of this method.
Proposition 2 proof raises a question: what if there is a small scaling factor in front of $2/(LK)$? Can this fix the issue of non-convergence? Moreover, I do not see how it implies the conclusion of Remark 1 that a step size has to be $\eta \leq \mathcal{O}(1/K)$.

**W3.** Validity of the interpolation condition in practice is questionable. It restricts the local gradients' values at the optimum but does not necessarily correspond to near-zero values of the local losses, which seems to be observed in experiments. It's also unclear if the Strong Growth Condition (SGC) is realistic for distributed deep neural network training. The paper presents these assumptions as more suitable for such problem setting. However, their validity is not explored in experiments.

**W4.** The experimental results display a multi-phase behavior of training loss reciprocal not anticipated or explained by the provided theoretical findings. The authors claim that their theory shows improvements coming from using local steps. However, Figure 1a clearly shows that in the first phase, it is not really the case.

The second experimental results implication seems to go against a common wisdom of reducing the number of local steps throughout optimization. The claim
> enforce more frequent communication at ﬁrst

contradicts the results of Figure 1a, which shows that doing more than 1 local step does not bring improvements.

## Questions
**Q1.** Assumption 1 is a stronger version of the standard (strong-)convexity typically used in distributed optimization. It requires the inequality to hold for any realization of random variables $\xi_i$. Why this modification is used? Is it necessary for obtaining the results, or it can be done for a more standard assumption as well?
The same question holds for Assumption 2. If $\mathcal{D}_i$ is a uniform distribution typical for distributed learning settings leading to a finite-sum scenario, then the interpolation condition has to hold for every single local data sample and not just for the average local loss $f_i$.

**Q2.** Can the presented results be obtained as a special case from a more general analysis of a non-overparametrized setting?

**Q3.** Why the data was distributed across clients exactly in this way?

**Q4.** Why are the convergence curves in Figure 4 of the Appendix truncated prematurely for smaller numbers of local steps?

### Minor

1. The paper has non-functioning links and references, and the outline isn't displayed properly. Local SGD, referred to as Algorithm 1, is incorrectly labeled as Algorithm 0.

2. > Furthermore, it was shown that a mini-batch size larger than some threshold $m^*$ is essentially helpless for SGD

Please support this claim with a direct reference. Ideally, with a mathematical explanation as well.

3. It is recommended to add details on the step size tuning, as the current description limits reproducibility.

4. Paper organization is unconventional with the Conclusion followed by Theorems’ proofs, which are typically put together in the Appendix. If the authors want to highlight a novel analysis technique, it would be helpful to provide a proof sketch with core insights.

5. The abstract’s claim that Local SGD is analyzed in a *“heterogeneous data setting”* is misleading, given that the interpolation condition actually restricts data heterogeneity as it characterizes not only the model properties but the local data distributions as well.

#### Typos

End of page 4: nodesâĂŹ

Page 6, line 2: “le” -> \leq

___

Nesterov, Y. (2018). Lectures on Convex Optimization, volume 137 of Springer Optimization and Its Appli-
cations. Springer, Cham.

---

> ### Author Response · Authors · 2024-02-26
> **Thank You! 1/4**
>
> We thank the reviewer for the careful reading of our paper and the insightful comments.  In the following, we have addressed each of your comments separately.
>
> **Weaknesses**
>
> - **W1** *The paper falls short in demonstrating a decisive improvement in performance through the use of local steps, akin to previous studies on Local SGD. The convergence bounds are optimized for the minimum $K$
> , leaving ambiguity about the additional methodological insights derived from this research. Besides, the discussion after Theorem 1 says that ""local steps can speed up the convergence of Local SGD by at least factor 2"" However, the explanation provided is unclear. It does not say how exactly the number of local steps affects convergence. Theorem 2 doesn’t demonstrate any advantage of using local steps, showing results typical for local methods, which necessitate scaling the step size inversely to the number of local steps. I am curious if this result improves upon previous studies. However, the conclusion says that ""Moreover, we validated the eﬀectiveness of local steps in speeding up the convergence of Local SGD in various settings both theoretically and using extensive simulations."" which seems to be not fully supported.*
>
> RE:  We thank the reviewer for raising these points. First of all, we have added a footnote to elaborate on ""local steps can speed up the convergence of Local SGD by at least factor 2"":
>
> ""Similar to Woodworth et al. (2020a), we compare the convergence rate of Local SGD to Minibatch SGD with $R = T/K$ steps and a batch size $K$ times larger than that of Local SGD. The convergence rate of Minibatch SGD (for over-parameterized setting), as stated in Theorem 6 in Vaswani et al. (2019), is $\frac{4LK(1+\rho)\|\mathbf{x}^{(0)} -\mathbf{x}^*\|^2}{T}$, which is at least $4$ times slower than our rate when $c=0$. On the other hand, according to our analysis (using Lemma 2), we can show the convergence rate of Minibatch SGD as $\frac{2LK\|\mathbf{x}^{(0)} -\mathbf{x}^*\|^2}{T}$, which is $2$ times slower than Local SGD.""
>
> Second, we acknowledge that Theorem 2 does not demonstrate the advantage of using local steps, and we have added a sentence in the conclusion section to clarify our contribution: ""However, our theoretical results fall short in explaining the effectiveness of local steps in the later phase of training non-convex over-parameterized Neural Networks, as observed in our experiments. We leave this important issue as a future research direction.""
>
> Finally, we want to emphasize that the result of Theorem 2 does improve upon previous studies by providing an improved rate of $\mathcal{O}(\frac{K}{T})$ instead of $\mathcal{O}(\frac{1}{\sqrt{nT}})$, which confirms well with the simulation results in Phase 1, Figure 3(a) (and it is **not** a special case from a more general analysis).
>
>
> - **W2** *The lower bound results rely on a highly specific set of parameters $n,R$, limiting their general applicability. It is known that Gradient Descent can provably diverge for step size $\eta>1/L$ (Nesterov, 2018). Thus, it would be odd to expect a different result for a variation of this method. Proposition 2 proof raises a question: what if there is a small scaling factor in front of $2/LK$? Can this fix the issue of non-convergence? Moreover, I do not see how it implies the conclusion of Remark 1 that a step size has to be $\eta\le\mathcal{O}(1/K)$.*
>
> RE: We thank the reviewer for raising these points. First, we want to clarify that $\eta\le 1/L$ is not our result but, on the contrary, is what we take as a standard requirement. In other words, we used the fact that SGD can provably diverge for step size $\eta>1/L$ to derive our result that ``the convergence rates shown in Section 3 are indeed tight up to a constant factor." We have revised the first paragraph of Section 4 to make it clear.
>
> Second, we want to emphasize that we are providing two specific problem instances to establish a lower bound for the **theoretical** convergence rate of Local SGD under the over-parameterized setting and do not expect them to generalize. Since the theoretical convergence rate of Local SGD should apply to **all** problem instances, two specific problem instances suffice to show that no valid theoretical convergence rate can be established that is faster than $\mathcal{O}(K/T)$. By lower bound, we do not mean that Local SGD can not converge faster than $\mathcal{O}(K/T)$, which can actually happen quite often in practice.

---

> ### Author Response · Authors · 2024-02-26
> **Thank You! 2/4**
>
> - **W3** *Validity of the interpolation condition in practice is questionable. It restricts the local gradients' values at the optimum but does not necessarily correspond to near-zero values of the local losses, which seems to be observed in experiments. It's also unclear if the Strong Growth Condition (SGC) is realistic for distributed deep neural network training. The paper presents these assumptions as more suitable for such problem setting. However, their validity is not explored in experiments.*
>
> RE: We thank the reviewer for raising this point. We have performed additional experimental verification of Assumption 3 (SGC) in Appendix B. Specifically, we use the same problem setup as in Section 4.1 and use centralized SGD to train the aforementioned ResNet18 neural network on the Cifar10 dataset for a total of 1000 epochs. We plot the global gradient norm and the maximum of per-sample gradient norm of the model, as well as the ratio between the two gradient norms. The results show that Assumption 3 (SGC) is indeed a valid assumption for over-parameterized models in practice. Since Assumption 3 (SGC) implies Assumption 2 (Interpolation), this is also evidence that Assumption 2 (Interpolation) is a valid assumption for over-parameterized models in practice.
>
> - **W4** *The experimental results display a multi-phase behavior of training loss reciprocal not anticipated or explained by the provided theoretical findings. The authors claim that their theory shows improvements coming from using local steps. However, Figure 1a clearly shows that in the first phase, it is not really the case. The second experimental results implication seems to go against a common wisdom of reducing the number of local steps throughout optimization. The claim ""enforce more frequent communication at ﬁrst"" contradicts the results of Figure 1a, which shows that doing more than 1 local step does not bring improvements.*
>
> RE: We thank the reviewer for raising this point. We want to point out that Phase 2 in the experimental results clearly shows that using local steps improves the convergence rate (however, our theoretical results fall short in explaining this, as we mentioned before), while Phase 1 in the experimental results shows a convergence rate of $\mathcal{O}(K/T)$ that confirms with Theorem 2 well (where local steps do not exhibit any help). We also want to point out that the claim does not contradict Figure 1a as "enforce more frequent communication at ﬁrst" means "use fewer local steps in Phase 1 (when local step does not bring improvements) but use more local steps in Phase 2 (when local step does bring improvements)."
>
> **Questions**
>
> - **Q1** *Assumption 1 is a stronger version of the standard (strong-)convexity typically used in distributed optimization. It requires the inequality to hold for any realization of random variables $\xi_i$. Why this modification is used? Is it necessary for obtaining the results, or it can be done for a more standard assumption as well? The same question holds for Assumption 2. If $\mathcal{D}_i$ is a uniform distribution typical for distributed learning settings leading to a finite-sum scenario, then the interpolation condition has to hold for every single local data sample and not just for the average local loss $f_i$.*
>
> RE: We thank the reviewer for raising these points. We need convexity to hold for any realization of random variables because we need to use inequality (8) in Proposition 3 for every $f_i(\mathbf{x}_i,\xi_i)$; this is essential for our analysis. While this is stronger than the standard convexity assumption, it is commonly satisfied in the empirical risk minimization problem, which is the main focus of training ML models, and it is also assumed in other works such as Ma et al. (2018). Assumption 2 also needs to hold for any realization of random variables. Interpolation essentially means that the empirical loss of every data point can be driven to zero. We have revised our manuscript to clarify this point.

---

> ### Author Response · Authors · 2024-02-26
> **Thank You! 3/4**
>
> - **Q2** *Can the presented results be obtained as a special case from a more general analysis of a non-overparametrized setting?*
>
> RE: We thank the reviewer for raising this point. No, they cannot. We have added a paragraph in Section 1.2 Contributions and Organization to elaborate on the technical novelty of our work:
>
> ""Our analysis built upon the techniques used by Ma et al. 2018 and Vaswani et al. 2019 for analyzing centralized SGD in the over-parameterized setting and applied them to analyze both the local descent progress and the global descent progress in Local SGD. Specifically, we adopt new techniques in the proof of Theorem 1 that directly relate the local progress with the global progress instead of measuring the progress made by $\bar{\mathbf{x}}^t$, where we made use of the consensus error, $i.e.$, $\frac{1}{n}\sum_{i=1}^n\mathbb{E}\|\mathbf{x}_i^t-\bar{\mathbf{x}}^t\|^2$ to *improve* convergence, which is in contrast to prior works. The new technique allows us to use a constant stepsize that does not scale with $\frac{1}{K}$ and to establish better bounds. In the proof of Theorem 2, we use the techniques of Ma et al. 2018 and Vaswani et al. 2019 to bound both the global descent progress and the consensus error of Local SGD. These techniques may be of independent interest to the readers.  ""
>
> - **Q3** *Why the data was distributed across clients exactly in this way?*
>
> RE: We thank the reviewer for raising this point. The data distribution in Section 4.1 follows a standard data partitioning scheme for the heterogeneous data setting, as in the seminal work by McMahan et al., 2017. In Section 4.2, we partition the dataset in three different ways to reflect different data similarity regimes.
>
> - **Q4** *Why are the convergence curves in Figure 4 of the Appendix truncated prematurely for smaller numbers of local steps?*
>
> RE: We thank the reviewer for raising this point. We have revised the ""Perceptron for Linearly Separable Dataset"" experiment in the Appendix and added it to Section 4.2. The experimental results confirm Theorem 1 very well and serve as strong support for our work. For the stopping criteria, we stop the algorithm after at most $10^6$ communication rounds or if the training loss is below $10^{−4}$.
>
> **Minor**
>
> - **1** *The paper has non-functioning links and references, and the outline isn't displayed properly. Local SGD, referred to as Algorithm 1, is incorrectly labeled as Algorithm 0.*
>
> RE: We thank the reviewer for raising these points. We have corrected these typos.
>
> - **2** *""Furthermore, it was shown that a mini-batch size larger than some threshold $m^*$ is essentially helpless for SGD"" Please support this claim with a direct reference. Ideally, with a mathematical explanation as well.*
>
> RE: We thank the reviewer for raising this point. We have clarified this point as:
>
> ""Furthermore, it was shown by Ma et al. (2018) that under certain conditions, a mini-batch size larger than some threshold $m^*$ is essentially helpless for SGD.""
>
> - **3** *It is recommended to add details on the step size tuning, as the current description limits reproducibility.*
>
> RE: We thank the reviewer for raising this point. We have specified that we choose $\eta = 0.1$ in Section 4.1 and $\eta = 0.075$ in Section 4.2.
>
> - **4** *Paper organization is unconventional with the Conclusion followed by Theorems’ proofs, which are typically put together in the Appendix. If the authors want to highlight a novel analysis technique, it would be helpful to provide a proof sketch with core insights.*
>
> RE: Thank you. We have put the proofs in Appendix A.
>
> - **5** *The abstract’s claim that Local SGD is analyzed in a ""heterogeneous data setting"" is misleading, given that the interpolation condition actually restricts data heterogeneity as it characterizes not only the model properties but the local data distributions as well.*
>
> RE: We thank the reviewer for raising this point. The ""heterogeneous data setting"" is in contrast to the ""iid data setting"", where it is assumed that the data distributions across all nodes are iid, i.e., $f_i(\mathbf{x}) = f(\mathbf{x}),\forall i$. The terminology is standard in the federated learning community. It is also common that some restrictions of data similarity are assumed under the heterogeneous data setting, e.g., Karimireddy et al., 2020b assumed Bounded Gradient Dissimilarity, and Koloskova et al., 2020 assumed Bounded Noise.
>
> **Typos**
>
> *End of page 4: nodesâĂŹ; Page 6, line 2: “le” -> \leq*
>
> RE: We thank the reviewer for raising these points. We have corrected these typos.

---

> ### Author Response · Authors · 2024-02-26
> **Thank You! 4/4**
>
> **Requested Changes**
>
> - **C1** *The paper should clarify how its analysis differs from prior works, elaborating on the technical novelties and what it is based upon.*
>
> RE: We thank the reviewer for raising this point. We have added a paragraph in Section 1.2 Contributions and Organization to elaborate on the technical novelty of our work:
>
> ""Our analysis built upon the techniques used by Ma et al. 2018 and Vaswani et al. 2019 for analyzing centralized SGD in the over-parameterized setting and applied them to analyze both the local descent progress and the global descent progress in Local SGD. Specifically, we adopt new techniques in the proof of Theorem 1 that directly relate the local progress with the global progress instead of measuring the progress made by $\bar{\mathbf{x}}^t$, where we made use of the consensus error, $i.e.$, $\frac{1}{n}\sum_{i=1}^n\mathbb{E}\|\mathbf{x}_i^t-\bar{\mathbf{x}}^t\|^2$ to *improve* convergence, which is in contrast to prior works. The new technique allows us to use a constant stepsize that does not scale with $\frac{1}{K}$ and to establish better bounds. In the proof of Theorem 2, we use the techniques of Ma et al. 2018 and Vaswani et al. 2019 to bound both the global descent progress and the consensus error of Local SGD. These techniques may be of independent interest to the readers.  ""
>
> - **C2** *Definition 1 appears non-standard in the context of distributed learning literature. A more comprehensive discussion on this, particularly if it acts as an additional assumption on local losses $f_i$, would strengthen the paper.*
>
> RE: We thank the reviewer for raising this point. It is **not** an additional assumption on local losses, as we already pointed out in the paper that inequality (5) is **always** satisfied with $c=0$. We acknowledge that a more comprehensive discussion on what conditions will be sufficient to allow $c>0$ would be very valuable. However, this is beyond the scope of this work, and we plan to explore it in our future work.
>
>
> Once again, we would like to thank you for all the insightful comments, as they were very helpful in improving our paper, and we hope that our changes will be found satisfactory.

---

### Review · Reviewer_NakU · 2024-01-07

**Summary Of Contributions:**

Main Arguments:
- Modern machine learning models are often over-parameterized and can interpolate the data by driving the empirical loss to zero.
- Existing analysis of Local SGD (FedAvg) convergence does not fully explain its faster convergence compared to Minibatch SGD in practice when training large neural networks.
- By assuming interpolation, the paper provides tighter convergence rates for Local SGD that depend on the total number of iterations T (compared to existing results depending on number of nodes n).
- This partially explains the faster convergence of Local SGD and effectiveness of local steps when training large neural networks.

Contributions:
- For general convex functions, proved error bounds of O(1/T) under a mild data similarity assumption and O(K/T) otherwise (improving on best previous O(1/sqrt(nT))).
- For non-convex functions, proved an error bound of O(K/T) (improving on best previous O(1/sqrt(nT))).
- Provided problem instances showing the analysis is tight up to constants for small stepsizes.
- Validated theory with large-scale experiments on deep neural networks exhibiting the O(1/T) convergence.

Key Findings:
- Local steps can speed up Local SGD by a factor of at least 2 even in the worst case.
- Slight similarity in local functions is critical for the fast O(1/T) convergence rate.
- Experiments show two-phase convergence: O(K/T) initially and O(1/T) later when close to optimum.

**Audience:**

Yes

**Broader Impact Concerns:**

This work is primarily theoretical in nature. Hence, there is no identifiable potential for negative societal impact arising from this work.

**Claims And Evidence:**

Yes

**Requested Changes:**

Please fix typos in the following places:
- Theorem 2 stepsize bound;
- Page 4: There is a typo after the phrase "The server then computes the average of all nodes...".

**Strengths And Weaknesses:**

Strengths:
- Rigorous theoretical analysis with clear assumptions.
- Tight lower bounds confirming analysis.
- Extensive experiments on deep nets validating theory.

No major weaknesses were found.

---

> ### Author Response · Authors · 2024-02-26
> **Thank You!**
>
> We thank the reviewer for the careful reading of our paper and the positive review. We have revised our manuscript according to your suggestions.
>
> **Requested Changes:**
>
> - *Theorem 2 stepsize bound; Page 4: There is a typo after the phrase "The server then computes the average of all nodes...".*
>
> RE: We thank the reviewer for raising these points. We have corrected these typos.
>
> Once again, we thank you for this positive comment, and we hope our changes will be satisfactory.

---

### Review · Reviewer_SVJV · 2024-02-14

**Summary Of Contributions:**

The paper establishes new convergence results for local SGD. The results are derived under over-parameterized setup. The convergence rates improve upon the existing best known rates for convex and nonconvex loss functions. Some problem instances are given to imply the results are tight.  Numerical experiments validate the results.

**Audience:**

Yes

**Claims And Evidence:**

Yes

**Requested Changes:**

1. The comparison with convergence results between local SGD and SGD in over-parameterized setting is insufficient. It particular, how the analysis, assumptions and results differ from (Ma et al 2018, Vaswani et al 2019).

2. Because the main contributions of the paper is the improved convergence rate, it would be better to have a table summarizing the results, along with the existing works. The table shall compare the convergence bounds and assumptions.

3. More discussions on the assumptions are needed. In particular, when Assumption 2 and 3 are satisfied. The paper referred to (Vaswani et al. 2019) for examples, which is unclear and insufficient.

Other minor comments:

1. The assumption that f is lower bounded, L smooth and stochastic gradient is unbiased should be put in Assumption block.

2. Page 4 Section 3 typo: nodes

3. In Algorithm 1, step 4 should be nabla f_i? Also, Step 7 should be sum_j x_j? Step 6-10 should be moved outside the loop of update?

4. In Theorem 1, should be algorithm 1.

**Strengths And Weaknesses:**

*Strength*: The paper derives novel convergence bounds for local SGD, which improves general understanding of local SGD performance in over-parameterized setting.

*Weaknesses*: see below.

---

> ### Author Response · Authors · 2024-02-26
> **Thank You!**
>
> We thank the reviewer for careful reading of our paper and the insightful comments.  In the following, each of your points has been dealt individually and we respond them accordingly.
>
> **Requested Changes:**
>
> - *The comparison with convergence results between local SGD and SGD in over-parameterized setting is insufficient. It particular, how the analysis, assumptions and results differ from (Ma et al 2018, Vaswani et al 2019).*
>
> RE: We thank the reviewer for raising this point. Both the works by Ma et al. 2018 and Vaswani et al. 2019 only deal with centralized SGD, while we work on Local SGD, whose analysis bears the complicated nature of distributed optimization, including the intricate interplay of the local descent progress, the global descent progress, and the consensus error. We have added a paragraph in Section 1.2 Contributions and Organization to elaborate on the technical novelty of our work:
>
> ``Our analysis built upon the techniques used by Ma et al. 2018 and Vaswani et al. 2019 for analyzing centralized SGD in the over-parameterized setting and applied them to analyze both the local descent progress and the global descent progress in Local SGD. Specifically, we adopt new techniques in the proof of Theorem 1 that directly relate the local progress with the global progress instead of measuring the progress made by $\bar{\mathbf{x}}^t$, where we made use of the consensus error, $i.e.$, $\frac{1}{n}\sum_{i=1}^n\mathbb{E}\|\mathbf{x}_i^t-\bar{\mathbf{x}}^t\|^2$ to *improve* convergence, which is in contrast to prior works. The new technique allows us to use a constant stepsize that does not scale with $\frac{1}{K}$ and to establish better bounds. In the proof of Theorem 2, we use the techniques of Ma et al. 2018 and Vaswani et al. 2019 to bound both the global descent progress and the consensus error of Local SGD. These techniques may be of independent interest to the readers.  "
>
> - *Because the main contributions of the paper is the improved convergence rate, it would be better to have a table summarizing the results, along with the existing works. The table shall compare the convergence bounds and assumptions.*
>
> RE: We thank the reviewer for raising this point. We have added Table 1 on page 3, which summarizes the comparison between the convergence bounds and assumptions of our work and the existing works.
>
> - *More discussions on the assumptions are needed. In particular, when Assumption 2 and 3 are satisfied. The paper referred to (Vaswani et al. 2019) for examples, which is unclear and insufficient.*
>
> RE: We thank the reviewer for raising this point. We have performed additional experimental verification of Assumption 3 (SGC) in Appendix B. Specifically, we use the same problem setup as in Section 4.1 and use centralized SGD to train the aforementioned ResNet18 neural network on the Cifar10 dataset for a total of 1000 epochs. We plot the global gradient norm and the maximum of per-sample gradient norm of the model as well as the ratio between the two gradient norms. The results show that Assumption 3 (SGC) is indeed a valid assumption for over-parameterized models in practice. Since Assumption 3 (SGC) implies Assumption 2 (Interpolation), this is also evidence that Assumption 2 (Interpolation) is a valid assumption for over-parameterized models in practice.
>
> **Minor comments:**
>
> - *The assumption that f is lower bounded, L smooth and stochastic gradient is unbiased should be put in Assumption block.*
>
> RE: Thank you. We have put it in an Assumption block.
>
> - *Page 4 Section 3 typo: nodes.  In Algorithm 1, step 4 should be nabla f_i? Also, Step 7 should be sum_j x_j? Step 6-10 should be moved outside the loop of update?  In Theorem 1, should be algorithm 1.*
>
> RE: We thank the reviewer for raising these points. We have corrected these typos (except for the fact that Steps 6-10 are indeed inside the loop of the update).
>
>
>
> We would like to thank you for all the insightful comments, as they were very helpful in improving our paper, and we hope that these changes will be found satisfactory.

---

### Author Response · Authors · 2024-02-26
**Thank you!**

We thank the editor for managing the review of our paper. We thank the reviewers for the comments they provided, we are sure including their suggestions improved the quality of our paper. Now, we are submitting a revised version of the manuscript that address the concerns raised by the reviewers.

In the edited version of our manuscript the changes and comments incorporated in the text are identified in color blue.

We hope the current version of the manuscript provides satisfactory responses to the reviewers' concerns. Please contact us with further doubts or questions about our response.

---

### Decision · Action_Editor_Ab6U · 2024-03-21

**Recommendation:** Accept with minor revision

**Comment:**

The main contributions are the effect of overparameterization in improving the convergence of local SGD. While the effect of overparameterization has been known for SGD, its power in accelerating the convergence has not been studied for local SGD. For both convex and nonconvex problems, the paper shows that local SGD in overparameterization setting achieves the rate $O(K/T)$, which are faster than the existing rate $O(1/\sqrt{nT})$ without overparameterization ($K$ is the number of local steps, $T$ is the total number of iterations and $n$ is the number of nodes). Furthermore, the paper derives rate $O(1/T)$ under a mild data similarity assumption. The paper also presents two problem instances, which show that the derived convergence rates are tight up to constant factors. These results are interesting to understand why local SGD achieves impressive performance to train overparameterized models. All the reviewers appreciate the theoretical contributions.

Minor comment:

Abstract ", These bounds improve" should be ". These bounds improve"

**Audience:**

Local SGD is an efficient distributed optimization algorithm with wide applications. The theoretical understanding on the convergence of SGD is important. The paper presents interesting results on the convergence of local SGD for interpolation, and should be interesting to many audience in the optimization and machine learning community.

**Claims And Evidence:**

The paper studies the convergence of local SGD in an overparameterization setting where the model can interpolate the training examples. The paper presents convergence rates for local SGD applied to convex, nonconvex problems and shows how the interpolation improves the convergence rates. These claims are supported by both rigorous theoretical analysis and empirical analysis on large-scale datasets. For example, the paper applies local SGD to train an over-parameterized ResNet18 neural network on the Cifar10 dataset, and the experimental results reveal the convergence behavior of Local SGD.

---

> ### Author Response · Authors · 2024-03-30
> **Response to Action Editor Ab6U**
>
> Thank you for managing the reviews of our work and the helpful feedback! We have submitted the camera ready version with the typo fixed.